# MAST: A Sparse Training Framework for Multi-agent Reinforcement Learning

## Abstract

Deep Multi-agent Reinforcement Learning (MARL) is often confronted with large state and action spaces, necessitating the utilization of neural networks with extensive parameters and incurring substantial computational overhead. Consequently, there arises a pronounced need for methods that expedite training and enable model compression in MARL. Nevertheless, existing training acceleration techniques are primarily tailored for single-agent scenarios, as the task of compressing MARL agents within sparse models presents unique and intricate challenges. In this paper, we introduce an innovative **M**ulti-**A**gent **S**parse **T**raining (MAST) framework. MAST capitalizes on gradient-based topology evolution to exclusively train multiple MARL agents using sparse networks. This is then combined with a novel hybrid TD-$(\lambda)$ schema, coupled with the Soft Mellowmax Operator, to establish dependable learning targets, particularly in sparse scenarios. Additionally, we employ a dual replay buffer mechanism to enhance policy stability within sparse networks. Remarkably, our comprehensive experimental investigation on the SMAC benchmarks, for the first time, that deep multi-agent Q learning algorithms manifest significant redundancy in terms of Floating Point Operations (FLOPs). This redundancy translates into up to 20-fold reduction in FLOPs for both training and inference, accompanied by a commensurate level of model compression, all achieved with less than $3\%$ performance degradation.

## 1 Introduction

Multi-agent reinforcement learning (MARL) (Shoham & Leyton-Brown, 2008), combined with deep neural networks, has not only revolutionized the field of artificial intelligence but also demonstrated remarkable success across a diverse spectrum of critical applications. From conquering multi-agent video games like Quake III Arena (Jaderberg et al., 2019), StarCraft II (Mathieu et al., 2021), Dota 2 (Berner et al., 2019), and Hide and Seek (Baker et al., 2019) to guiding autonomous robots through intricate real-world environments (Shalev-Shwartz et al., 2016; Da Silva et al., 2017; Chen et al., 2020b), deep MARL has established itself as an indispensable and versatile tool for addressing complex, multifaceted challenges. Its unique ability to capture intricate interactions and dependencies among multiple agents has generated novel insights and solutions, solidifying its role as a transformative paradigm across various domains (Zhang et al., 2021; Albrecht et al., 2023).

Nonetheless, the extraordinary success of deep MARL comes at a substantial computational cost. Training these agents involves the intricate task of adapting neural networks to accommodate an expanded parameter space, especially when the number of agents involved is substantial. For example, the training regimen for AlphaStar (Mathieu et al., 2021), designed for StarCraft II, which spanned an arduous 14-day period, utilizing 16 TPUs per agent. The OpenAI Five (Berner et al., 2019) model for Dota 2 underwent a marathon training cycle, spanning 180 days and tapping into thousands of GPUs. This exponential growth in computational demands as the number of agents increases presents a formidable challenge when deploying MARL in practical problems. The joint action and state spaces swell exponentially, imposing a steep demand on computational resources.

Researchers have explored dynamic sparse training (DST) like SET (Mocanu et al., 2018) and RigL (Evci et al., 2020) to address computational challenges. While initial attempts at sparse single-agent deep reinforcement learning (DRL) training have been made in (Sokar et al., 2022; Graesser et al., 2022), DST methods have struggled to achieve consistent model compression across diverse environments. RLx2 (Tan et al., 2022) enables sparse neural network training for DRL but is ineffective

for multi-agent RL (MARL). In a motivating experiment, we tested various sparse training methods on the `3s5z` tasks from SMAC (Samvelyan et al., 2019) using a neural network with only 10% of its original parameters, as shown in Figure 1. Classical DST methods, including SET and RigL, as well as RLx2 for single-agent RL, perform poorly in MARL scenarios, not to mention static sparse networks (SS). In contrast, our MAST framework achieves over 90% win rate. The sole prior effort to train sparse MARL agents, as in (Yang et al., 2022), prunes agent networks during training with weight grouping (Wang et al., 2019). However, this approach fails to maintain sparsity throughout training, reaching only about 80% sparsity. Moreover, their experimental evaluation is confined to a two-user environment, PredatorPrey-v2, in MuJoCo (Todorov et al., 2012).

These observations underscore the fact that, despite their promise, the application of sparse networks in the context of MARL remains largely uncharted territory. The existing state-of-the-art DST technique, RLx2 (Tan et al., 2022), while effective in single-agent scenarios, demonstrates limitations when confronted with the challenges posed by MARL. MARL introduces unique complexities, including larger system spaces, the non-stationarity inherent in multi-agent training, and the partially observable nature of each agent. Consequently, a critical and intriguing question emerges: *Can we train MARL agents using sparse networks throughout?*

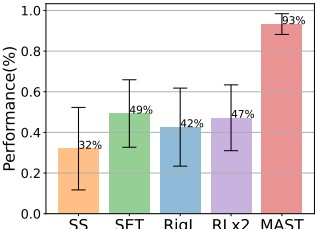

Figure 1: Comprison of different sparse traning methods.

We give an affirmative answer to the question by presenting a novel sparse training framework, MAST, tailored explicitly for value decomposition methods in MARL. It leverages gradient-based topology evolution, offering a powerful tool for the efficient exploration of network configurations in sparse models. Notably, our investigation has unveiled the formidable challenges faced by MARL algorithms in the realm of ultra-sparse models, i.e., inaccurate learning targets and training instability. To surmount these challenges, MAST introduces innovative solutions. We present a novel hybrid TD($\lambda$) target mechanism, coupled with the Soft Mellowmax operator, which facilitates precise value estimation even in the face of extreme sparsity. Additionally, MAST unveils a dual buffer mechanism designed to bolster training stability in sparse environments. As a result, MAST empowers the training of highly efficient MARL agents with minimal performance compromise, employing ultra-sparse networks throughout the training process. Our extensive experiments, conducted across several popular MARL algorithms, validate MAST's position at the forefront of sparse training. These experiments reveal MAST's ability to achieve model compression ratios ranging from $5\times$ to $20\times$, all while incurring minimal performance trade-offs, typically under $3\%$. Moreover, MAST boasts the impressive capability to reduce FLOPs required for both training and inference by up to an astounding $20\times$, showing a large margin over other baselines including SET (Mocanu et al., 2018), RigL (Evci et al., 2020) and RLx2 (Tan et al., 2022).

## 2 RELATED WORK

Sparse networks, initially proposed in deep supervised learning, can train a 90%-sparse network without performance degradation from scratch. However, for deep reinforcement learning, the learning target is not fixed but evolves in a bootstrap way (Tesauro et al., 1995), and the distribution of the training data can also be non-stationary (Desai et al., 2019), which makes the sparse training more difficult. We list some representative works for training sparse models from supervised learning to reinforcement learning. A more comprehensive illustration can be found in Appendix A.1.

**Sparse Models in Supervised Learning**  Various techniques have been explored for creating sparse networks, ranging from pruning pre-trained dense networks (Han et al., 2015; 2016; Srinivas et al., 2017), to employing derivatives (Dong et al., 2017; Molchanov et al., 2019). Another avenue of research revolves around the Lottery Ticket Hypothesis (LTH) (Frankle & Carbin, 2019), which posits the feasibility of training sparse networks from scratch, provided a sparse "winning ticket" initialization is identified. Additionally, there is a body of work dedicated to training sparse neural networks from the outset, involving techniques that evolve the structures of sparse networks during training. Examples include SET (Mocanu et al., 2018) and RigL (Evci et al., 2020).

**Sparse Models in Single-Agent RL**  Existing research (Schmitt et al., 2018; Zhang et al., 2019) has employed knowledge distillation with static data to ensure training stability and generate small dense agents. Policy Pruning and Shrinking (PoPs) (Livne & Cohen, 2020) generates sparse agents through iterative policy pruning. Another line of investigation aims to train sparse DRL models from scratch, eliminating the necessity of pre-training a dense teacher. (Sokar et al., 2022; Graesser et al.,

2022) utilize the DST in single-agent RL, achieving a $50\% - 80\%$ sparsity level. More recently, RLx2 (Tan et al., 2022) has demonstrated the capacity to train DRL agents with highly sparse neural networks from scratch, yet RLx2 performs poorly in MARL as demonstrated in Section 5.1.

**Sparse Models in MARL**  The existing endeavour has made attempts to train sparse MARL agents, such as (Yang et al., 2022), which prunes networks for multiple agents during training. Another avenue of research seeks to enhance the scalability of MARL through sparse architectural modifications. For instance, (Sun et al., 2020) uses a sparse communication graph with graph neural networks to reduce problem scale, and (Kim & Sung, 2023) adopts structured pruning for a deep neural network to extend the scalability. Others focus on parameter sharing between agents to reduce the number of trainable parameters, with representative works including (Li et al., 2021; Christianos et al., 2021). Yet existing methods fail to maintain high sparsity throughout the training process.

## 3 DEEP MULTI-AGENT REINFORCEMENT LEARNING PRELIMINARIES

We model the MARL problem as a decentralized partially observable Markov decision process (Oliehoek et al., 2016), represented by a tuple $\langle \mathcal{N}, \mathcal{S}, \mathcal{U}, P, r, \mathcal{Z}, O, \gamma \rangle$, with detailed specification in Appendix A.2. Deep Multi-Agent $Q$-learning extends the deep $Q$ learning method (Mnih et al., 2013) to multi-agent scenarios (Sunehag et al., 2018; Rashid et al., 2020b; Son et al., 2019). Each agent encounters partial observability, and the agent-wise Q function is defined over its history $\tau_i$ as $Q_i$ for agent $i$. Subsequently, the joint action-value function $Q_{\text{tot}}(\boldsymbol{\tau}, \boldsymbol{u})$ operates over the joint action-observation history $\boldsymbol{\tau}$ and joint action $\boldsymbol{u}$. The objective, given transitions $(\boldsymbol{\tau}, \boldsymbol{u}, r, \boldsymbol{\tau}')$ sampled from the experience replay buffer $\mathcal{B}$, is to minimize the mean squared error loss $\mathcal{L}(\theta)$ on the temporal-difference (TD) error $\delta = y - Q_{\text{tot}}(\boldsymbol{\tau}, \boldsymbol{u})$. Here, the TD target $y = r + \gamma \max_{\boldsymbol{u}'} \tilde{Q}_{\text{tot}}(\boldsymbol{\tau}', \boldsymbol{u}')$, where $\tilde{Q}_{\text{tot}}$ is the target network for the joint action $Q$-function, periodically copied from $Q_{\text{tot}}$. Parameters of $Q_{\text{tot}}$ are updated using $\theta' = \theta - \alpha \nabla_\theta \mathcal{L}(\theta)$, with $\alpha$ representing the learning rate.

**CTDE**  We focus on algorithms that adhere to the Centralized Training with Decentralized Execution (CTDE) paradigm (Oliehoek et al., 2008; Kraemer & Banerjee, 2016). Within this paradigm, agents undergo centralized training, where they have access to the complete action-observation history and global state information. However, during execution, they are constrained to their individual local action-observation histories. To efficiently implement CTDE, the Individual-Global-Maximum (IGM) property (Son et al., 2019), defined in Eq. (1), serves as a key mechanism.

$$\arg\max_{\boldsymbol{u}} Q_{\text{tot}}(s, \boldsymbol{u}) = \left( \arg\max_{u_1} Q_1(s, u_1), \cdots, \arg\max_{u_N} Q_N(s, u_N) \right). \tag{1}$$

Many deep MARL algorithms adhere to the IGM criterion, such as the QMIX series algorithms (Rashid et al., 2020b;a). These algorithms employ a mixing network $f_s$ with non-negative weights, enabling the joint Q-function to be expressed as $Q_{\text{tot}}(s, \boldsymbol{u}) = f_s(Q_1(s, u_1), \cdots, Q_N(s, u_N))$.

## 4 BOOSTING THE PERFORMANCE OF SPARSE MARL AGENTS

This section outlines the pivotal components of the MAST framework. Initially, MAST relies on the gradient-based topology evolution for finding proper sparse network topology. However, as depicted in Figure 1, training ultra-sparse MARL models using topology evolution presents substantial challenges. Consequently, MAST introduces innovative solutions to address the accuracy of value learning in ultra-sparse models by concurrently refining training data targets and distributions.

### 4.1 TOPOLOGY EVOLUTION

The topology evolution mechanism in MAST follows the RigL method (Evci et al., 2020). RigL improves the optimization of sparse neural networks by leveraging weight magnitude and gradient information to jointly optimize model parameters and connectivity. After setting the initial network sparsity, the initial sparsity distribution of each layer is decided by Erdős–Rényi strategy from (Mocanu et al., 2018). As shown in Figure 2,

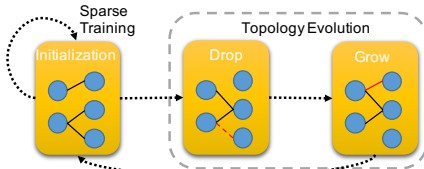

Figure 2: Network topology evolution.

RigL periodically dynamically drops a subset of existing connections with the smallest absolute weight values and concurrently grows an equivalent number of empty connections with the largest gradients. The diminishing update fraction $\zeta_t$ for connections follows $\zeta_t = \frac{\zeta_0}{2}(1 + \cos(\pi t/T_{\text{end}}))$, where $\zeta_0$ is the initial update fraction, and $T_{\text{end}}$ is the training steps. This process maintains the network sparsity throughout the training yet with strong evolutionary ability.

The topology evolution is detailed in Algorithm 1, where the symbol $\odot$ denotes the element-wise multiplication operator, while $M_\theta$ symbolizes the binary mask that delineates the sparse topology

for the network $\theta$. We set a low topology adjustment rate as prior studies (Evci et al., 2020; Tan et al., 2022), occurring at intervals of 200 gradient updates. This setup minimizes the computational burden of topology evolution, ensuring operational feasibility even on resource-constrained devices.

**Algorithm 1** Topology Evolution (Evci et al., 2020)

1: $\theta_l, N_l, s_l$: parameters, number of parameters, sparsity of layer $l$.
2: **for** each layer $l$ **do**
3:     $k = \zeta_t(1 - s_l)N_l$
4:     $\mathbb{I}_{\text{drop}} = \text{ArgTopK}(-|\theta_l \odot M_{\theta_l}|, k)$
5:     $\mathbb{I}_{\text{grow}} = \text{ArgTopK}_{i \notin \theta_l \odot M_{\theta_l} \setminus \mathbb{I}_{\text{drop}}}(|\nabla_{\theta_l} L, k|)$
6:     Update $M_{\theta_l}$ according to $\mathbb{I}_{\text{drop}}$ and $\mathbb{I}_{\text{grow}}$
7:     $\theta_l \leftarrow \theta_l \odot M_{\theta_l}$
8: **end for**

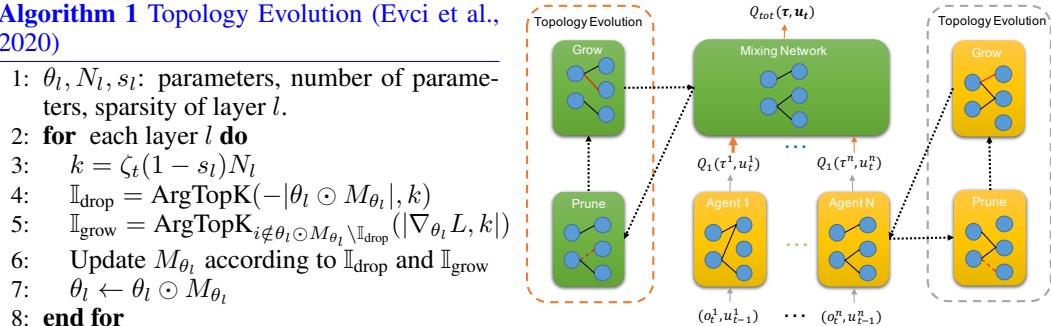

Figure 3: An overview of MAST-QMIX.

Figure 3 provides an overview of sparse models when MAST is applied to QMIX. MAST introduces three innovative solutions to achieve accurate value learning in ultra sparse models: i) Hybrid TD($\lambda$) targets to mitigate estimation errors from network sparsity. ii) The Soft Mellowmax operator to reduce overestimation in sparse models. iii) Dual replay buffers to stabilize sparse training.

### 4.2 HYBRID TD($\lambda$) TARGETS

In MAST, we utilize hybrid TD($\lambda$) targets to generate reliable learning targets, which achieves a good trade-off between sparse network fitting errors and learning variances. We will first introduce the benefit of TD($\lambda$) targets and then show the necessity of the hybrid scheme.

**TD($\lambda$) Targets** Temporal difference (TD) learning is a fundamental method for determining an optimal policy in reinforcement learning, with the value network iteratively updated by minimizing a squared loss driven by the TD target. Denote the multi-step return $\mathcal{T}_t^{(n)}$ at timestep $t$ for deep multi-agent Q learning as $\mathcal{T}_t^{(n)} = \sum_{i=t}^{t+n} \gamma^{i-t} r_i + \gamma^{n+1} \max_{\boldsymbol{u}} Q_{\text{tot}}(s_{i+n+1}, \boldsymbol{u})$. As evidenced in prior works (Sokar et al., 2022; Tan et al., 2022), sparse networks, denoted by parameters $\hat{\theta} = \theta \odot M_\theta$, where $\odot$ signifies element-wise multiplication, and $M_\theta$ is a binary mask representing the network's sparse topology, operates within a reduced hypothesis space with fewer parameters. Consequently, the sparse network $\hat{\theta}$ may induce a large bias, such that the learning targets become unreliable. Denote the network fitting error as $\epsilon(s, \boldsymbol{u}) = Q_{\text{tot}}(s, \boldsymbol{u}; \theta) - Q_{\text{tot}}^{\pi_t}(s, \boldsymbol{u})$, it will be larger under an improper sparsified model compared to a dense network, as evidenced in Figure 1 where improper sparsified models fail in learning good policy. Specifically, Eq. (2) from (Tan et al., 2022) characterises the expected error between the multi-step TD target $\mathcal{T}_t^{(n)}$ and the true Q-function $Q_{\pi_t}$ associated with the target policy $\pi_t$ conditioned on transitions from the behaviour policy $b_t$, reveals that introducing a multi-step return target discounts the network fitting error by a $\gamma^n$ factor.

$$\mathbb{E}_{b_t}[\mathcal{T}_t^{(n)}(s, \boldsymbol{u})] - Q_{\text{tot}}^{\pi_t}(s, \boldsymbol{u}) = \underbrace{\left(\mathbb{E}_{b_t}[\mathcal{T}_t^{(n)}(s, \boldsymbol{u})] - \mathbb{E}_{\pi_t}[\mathcal{T}_t^{(n)}(s, \boldsymbol{u})]\right)}_{\text{Policy inconsistency error}} + \gamma^n \underbrace{\mathbb{E}_\pi[\epsilon(s_{t+n}, \pi_t(\boldsymbol{u}_{t+n}))]}_{\text{Network fitting error}}.$$

(2)

Thus, employing a multi-step return $\mathcal{T}_t^{(n)}$ with a sufficiently large $n$, e.g., $\mathcal{T}_t^{(\infty)}$ or Monte Carlo methods (Sutton & Barto, 2018), effectively diminishes the network fitting error by a very small factor of $\gamma^n$ approaching 0 for $\gamma < 1$. However, the Monte Carlo method is susceptible to large variance, which implies that an optimal TD target shall be a multi-step return with a judiciously chosen $n$, striking a balance between network fitting error and variances. This motivates us to introduce the TD($\lambda$) target (Sutton & Barto, 2018) to achieve good trade-off: $\mathcal{T}_t^\lambda = (1 - \lambda) \sum_{n=1}^{\infty} \lambda^{n-1} \mathcal{T}_t^{(n)}$ for $\lambda \in [0, 1]$, which average all of the possible multi-step returns $\{\mathcal{T}_t^{(n)}\}_{n=1}^{\infty}$ into a single return by using a weight that decays exponentially, and is computationally efficient with episode-form data.

**Hybrid Scheme** Previous studies (Fedus et al., 2020; Tan et al., 2022) have highlighted that an immediate shift to multi-step targets can exacerbate policy inconsistency error in Eq. (2). Since the TD($\lambda$) target $\mathcal{T}_t^\lambda$ averages all potential multi-step returns $\{\mathcal{T}_t^{(n)}\}_{n=1}^{\infty}$, an immediate transition to this target may encounter similar issues. We adopt a hybrid strategy inspired by the delayed approach in Tan et al. (2022). Initially, when the training step is less than $T_0$, we use one-step TD targets

$(\mathcal{T}_t^{(1)})$ to minimize policy inconsistency errors. As training progresses and the policy stabilizes, we seamlessly transition to TD($\lambda$) targets to mitigate sparse network fitting errors. Such a hybrid TD($\lambda$) mechanism ensures consistent and reliable learning targets, even within sparse models.

Furthermore, we empirically demonstrate the effectiveness of our proposed hybrid TD($\lambda$) targets on the `3s5z` task in the SMAC, as illustrated in Figure 4. Our findings underscore the pivotal role of TD($\lambda$) in enhancing the learning process of sparse models. Interestingly, we observe that including a 1-step return target during initial training, although slightly reducing sample efficiency, contributes significantly to the agents' learning in the final stages. This highlights the necessity of our hybrid approach for sparse networks. Moreover, we examine hybrid multi-step TD targets in RLx2 (Tan et al., 2022) for single-agent sparse training with a fixed $n = 3$, in our experiments on RigL-QMIX. Figure 4 clearly illustrates the superiority of our hybrid TD($\lambda$)

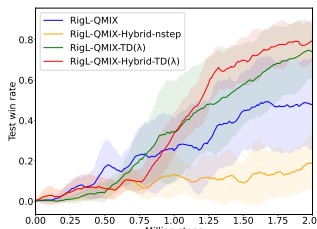

Figure 4: Performance comparison of various TD targets.

mechanism. This suggests the optimal TD target may not always be a fixed multi-step return; instead, an average value is a robust choice, coinciding with Figure 7.2 in (Sutton & Barto, 2018).

### 4.3 SOFT MELLOWMAX OPERATOR

We empirically observe that the overestimation issue still arises in sparse MARL models, significantly impacting performance. MAST utilizes a robust operator, i.e., Soft Mellomax operator from (Gan et al., 2021), to alleviate the overestimation and achieve accurate value estimation.

**Overestimation**  The max operator in the Bellman operator poses a well-known theoretical challenge, i.e., overestimation, hindering the convergence of various linear or non-linear approximation schemes (Tsitsiklis & Van Roy, 1996), which stands as a significant source of instability in the original deep Q-network (DQN) (Mnih et al., 2015). Deep MARL algorithms, including QMIX (Rashid et al., 2020b), also grapple with the overestimation issue. Recent research efforts (Gan et al., 2021; Pan et al., 2021) have aimed to alleviate overestimation through conservative operators and regularization techniques. Moreover, Our empirical investigations reveal that the overestimation issue persists in sparse models, significantly impacting performance.

Figure 5 illustrates the win rates and estimated values of QMIX with or without our Soft Mellowmax operator on `3s5z` in the SMAC. We derive estimated values by averaging over 40 episodes sampled from the replay buffer every $10,000$ timestep. Figure 5(a) shows that the performance of RigL-QMIX-SM outperforms RigL-QMIX, and Figure 5(b) shows that Soft Mellowmax operator

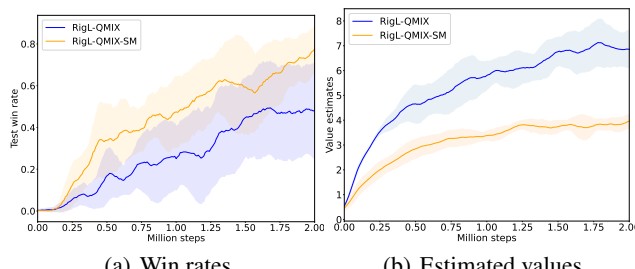

(a) Win rates  (b) Estimated values

Figure 5: Effects of Soft Mellowmax operator.

does effectively mitigate the overestimation bias. These emphasize that in sparse models, QMIX still faces overestimation issues, highlighting the critical importance of addressing overestimation.

**Soft Mellow operator**  For MARL algorithms satifying the IGM property in Eq. (1), we replace the max operator in $Q_i$ to Soft Mellowmax operator (Gan et al., 2021) in Eq. (3), to mitigate overestimation bias in the joint-action Q function within sparse models.

$$\text{sm}_\omega(Q_i(\tau, \cdot)) = \frac{1}{\omega} \log \left[ \sum_{u \in \mathcal{U}} \text{softmax}\left(Q_i(\tau, u)\right) \exp\left(\omega Q_i(\tau, u)\right) \right], \quad (3)$$

where $\text{softmax}_\alpha\left(Q_i(\tau, u)\right) = \frac{\exp(\alpha Q_i(\tau, u))}{\sum_{u' \in \mathcal{U}} \exp(\alpha Q_i(\tau, u'))}$, $\omega > 0$ and $\alpha \in \mathbb{R}$. Eq. (3) can be regarded as a specific instance of the weighted quasi-arithmetic mean (Beliakov et al., 2016). The $\text{softmax}_\alpha(Q)$ can be interpreted as a representation of policy probability, aligning with the framework of entropy regularization and KL divergence (Fox et al., 2015; Mei et al., 2019). Also note that when $\alpha = 0$, the Soft Mellowmax operator simplifies to the Mellomax operator $\text{mm}(\cdot)$ as:

$$\text{mm}_\omega(Q_i(\tau, \cdot)) = \frac{1}{\omega} \log \left[ \sum_{u \in \mathcal{U}} \frac{1}{|\mathcal{U}|} \exp\left(\omega Q_i(\tau, u)\right) \right]. \quad (4)$$

Also, $\lim_{\omega\to\infty} \text{mm}_\omega Q_i(\tau,\cdot) = \max_{\boldsymbol{u}} Q_i(\tau,u)$, $\lim_{\omega\to 0} \text{mm}_\omega Q_i(\tau,\cdot) = \frac{1}{|\mathcal{U}|}\sum_u Q_i(\tau,u)$ according to (Asadi & Littman, 2017). As demonstrated in (Gan et al., 2021), the Soft Mellomax operator extends the capabilities of the Mellomax operator in various aspects, including provable performance bounds, overestimation bias reduction, and sensitivity to parameter settings.

### 4.4 DUAL BUFFERS

Training with online data enhances learning stability but sacrifices sample (Song et al., 2023). Conversely, offline data training boosts sample efficiency at the expense of stability. Figure 6 displays training dynamics for RigL-QMIX and others in SMAC's `3s5z` task, revealing QMIX instability in sparse models. Inspired by (Li et al., 2022), MAST employs a hybrid approach with two replay buffers: $\mathcal{B}_1$ (offline, large capacity, typically around 5000) and $\mathcal{B}_2$ (online, smaller capacity, usually around 100). $\mathcal{B}_1$ follows an off-policy style, while $\mathcal{B}_2$ aligns with an on-policy style. In each step, MAST samples $b_1$ episodes from $\mathcal{B}_1$ and $b_2$ transitions from $\mathcal{B}_2$, conducting a gradient update based on a combined batch of size $(b_1 + b_2)$. As seen in Figure 6, dual buffers enhance QMIX's training stability under sparse models, leading to consistent policy improvements and higher rewards. This mechanism remains insensitive in dense cases where network parameters ensure stable policy improvements. Notably, while prior works have explored prioritized or dynamic-capacity buffers (Schaul et al., 2015; Tan et al., 2022), they may be not applicable here due to data being in episode form, since addressing partial observation issue in MARL using recurrent neural networks.

**Target Value and Loss Function** Combining hybrid TD($\lambda$) with the Soft Mellowmax operator, we modify the target $y$ as follows:

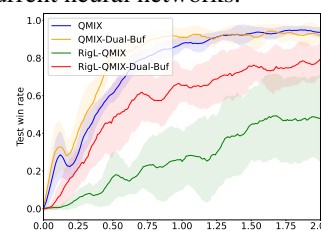

$$y_{\text{S}} = \begin{cases} G_t^{(1)}, & \text{if } t < T_0. \\ (1-\lambda)\sum_{n=1}^{\infty}\lambda^{n-1}\mathcal{T}_t^{(n)}, & \text{Otherwise.} \end{cases} \quad (5)$$

Here, $\lambda \in [0,1]$ is a hyperparameter, and $\mathcal{T}_t^{(n)} = \sum_{i=t}^{t+n}\gamma^{i-t}r_i + \gamma^{n+1}f_s\left(\text{sm}_\omega(\bar{Q}_1(\tau_1,\cdot)),\ldots,\text{sm}_\omega(\bar{Q}_N(\tau_N,\cdot))\right)$, where $f_s$ denotes the mixing network and $\bar{Q}_i$ is the target network of $Q_i$. The loss

Figure 6: Effects of dual buffers in QMIX.

function of MAST is defined as: $\mathcal{L}_{\text{S}}(\theta) = \mathbb{E}_{(s,\boldsymbol{u},r,s')\sim\mathcal{B}_1\cup\mathcal{B}_2}\left[\left(y_{\text{S}} - Q_{tot}(s,\boldsymbol{u})\right)^2\right]$. When $\lambda = 0$, it is equivalent to the 1-step TD target. When $\lambda = 1$, it can be thought of as the Monte Carlo method.

## 5 EXPERIMENTS

In this section, we conduct a comprehensive performance evaluation of MAST across four tasks: `3m`, `2s3z`, `3s5z`, and `2c_vs_64zg` from the SMAC benchmark (Samvelyan et al., 2019). MAST serves as a versatile sparse training framework specifically tailored for value decomposition-based Multi-Agent Reinforcement Learning (MARL) algorithms. In Section 5.1, we integrate MAST with state-of-the-art MARL algorithms, including QMIX (Rashid et al., 2020b), WQMIX (Rashid et al., 2020a), and RES (Pan et al., 2021), with detailed implementation given in Appendix A.3. This integration allows us to meticulously quantify the benefits derived from sparsification. To gain a profound understanding of the individual components that constitute MAST, we present a comprehensive ablation study in Section 5.2. Furthermore, we assess the performance of sparse models generated by MAST in Section 5.3. Detailed experimental configurations can be found in Appendix B. Also note that each reported result is based on the average performance over four independent runs, each utilizing distinct random seeds.

### 5.1 COMPARATIVE EVALUATION

Table 1 presents a comprehensive summary of our comparative evaluation, where MAST is benchmarked against the following baseline methods: (i) `Tiny`: Utilizing tiny dense networks with a parameter count matching that of the sparse model during training. (ii) `SS`: Employing static sparse networks with random initialization. (iii) `SET` (Mocanu et al., 2018): prunes connections based on their magnitude and randomly expands connections. (iv) `RigL` (Evci et al., 2020): This approach leverages dynamic sparse training, akin to MAST, by removing and adding connections based on magnitude and gradient criteria. (v) `RLx2` (Tan et al., 2022): A specialized dynamic sparse training framework tailored for single-agent reinforcement learning.

We set the same sparsity levels for both the joint Q function $Q_{\text{tot}}$, and each individual agent's Q function $Q_i$. For every algorithm and task, the sparsity level indicated in Table 1 corresponds to

the highest admissible sparsity threshold of MAST. Within this range, MAST's performance consistently remains within a $\pm 3\%$ margin compared to the dense counterpart, effectively representing the minimal sparse model size capable of achieving performance parity with the original dense model. All other baselines are evaluated under the same sparsity level as MAST. We assess the performance of each algorithm by computing the average win rate per episode over the final 20 policy evaluations conducted during training, with policy evaluations taking place at 10000-step intervals. Identical hyperparameters are employed across all 4 environments for 3 algorithms, detailed in Appendix B.3.

Table 1: Comparisons of MAST with sparse training baselines. Sp.: sparsity. Total Size: total model parameters (detailed in Appendix B.4). The data is all normalized w.r.t. the dense model.

| Alg. | Env. | Sp. | Total Size | FLOPs (Train) | FLOPs (Test) | Tiny (%) | SS (%) | SET (%) | RigL (%) | RLx2 (%) | Ours (%) |
|------|------|-----|-----------|---------------|--------------|----------|--------|---------|----------|----------|----------|
| Q-MIX | 3m | 95% | 0.066x | 0.051x | 0.050x | 98.3 | 91.6 | 96.0 | 95.3 | 12.1 | **100.9** |
|  | 2s3z | 95% | 0.062x | 0.051x | 0.050x | 83.7 | 73.0 | 77.6 | 68.2 | 45.8 | **98.0** |
|  | 3s5z | 90% | 0.109x | 0.101x | 0.100x | 68.2 | 34.0 | 52.3 | 45.2 | 50.1 | **99.0** |
|  | 64* | 90% | 0.106x | 0.100x | 0.100x | 58.2 | 40.2 | 67.1 | 48.7 | 9.9 | **96.4** |
|  | Avg. | 92% | 0.086x | 0.076x | 0.075x | 77.1 | 59.7 | 73.2 | 64.3 | 29.8 | **98.6** |
| WQ-MIX | 3m | 90% | 0.108x | 0.100x | 0.100x | 98.3 | 96.9 | 97.8 | 97.8 | 98.0 | **98.6** |
|  | 2s3z | 90% | 0.106x | 0.100x | 0.100x | 89.6 | 75.4 | 85.9 | 86.8 | 87.3 | **100.2** |
|  | 3s5z | 90% | 0.105x | 0.100x | 0.100x | 70.7 | 62.5 | 56.0 | 50.4 | 60.7 | **96.1** |
|  | 64* | 90% | 0.104x | 0.100x | 0.100x | 51.0 | 29.6 | 44.1 | 41.0 | 52.8 | **98.4** |
|  | Avg. | 90% | 0.106x | 0.100x | 0.100x | 77.4 | 66.1 | 70.9 | 69.0 | 74.7 | **98.1** |
| RES | 3m | 95% | 0.066x | 0.055x | 0.050x | 97.8 | 95.6 | 97.3 | 91.1 | 97.9 | **99.8** |
|  | 2s3z | 90% | 0.111x | 0.104x | 0.100x | 96.5 | 92.8 | 92.8 | 94.7 | 94.0 | **98.4** |
|  | 3s5z | 85% | 0.158x | 0.154x | 0.150x | 95.1 | 89.0 | 90.3 | 92.8 | 86.2 | **99.4** |
|  | 64* | 85% | 0.155x | 0.151x | 0.150x | 83.3 | 39.1 | 44.1 | 35.3 | 72.7 | **104.9** |
|  | Avg. | 89% | 0.122x | 0.116x | 0.112x | 93.2 | 79.1 | 81.1 | 78.5 | 87.7 | **100.6** |

**Performance** Table 1 unequivocally illustrates MAST's substantial performance superiority over all baseline methods in all four environments across the three algorithms. Notably, static sparse (`SS`) consistently exhibit the lowest performance on average, highlighting the difficulty of finding optimal sparse network topologies in the context of sparse MARL models. Dynamic sparse training methods, namely `SET` and `RigL`, slightly outperform (`SS`), although their performance remains unsatisfactory. Sparse networks also, on average, underperform tiny dense networks. However, MAST significantly outpaces all other baselines, indicating the successful realization of accurate value estimation through our MAST method,

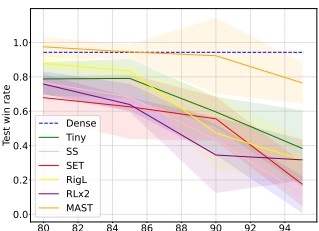

Figure 7: Performances under different sparsity.

which effectively guides gradient-based topology evolution. Notably, the single-agent method `RLx2` consistently delivers subpar results in all experiments, potentially due to its limited replay buffer capacity, severely hampering sample efficiency. To further substantiate the efficacy of MAST, we conduct performance comparisons across various sparsity levels in `3s5z`, as depicted in Figure 7. This reveals an intriguing observation: the performance of sparse models experiences a sharp decline beyond a critical sparsity threshold. Compared to conventional DST techniques, MAST significantly extends this critical sparsity threshold, enabling higher levels of sparsity while maintaining performance. Moreover, RES achieves a higher critical sparsity threshold than the other two algorithms with existing baselines, e.g., `SET` and `RigL`, achieving a sparsity level of over $80\%$ on average. However, it is essential to note that the Softmax operator in RES results in significantly higher computational FLOPs (as detailed in Appendix B.4.5), making it incomparable in terms of training and inference acceleration to MAST.

**FLOPs Reduction and Model Compression** In contrast to knowledge distillation or behavior cloning methodologies, exemplified by works such as (Livne & Cohen, 2020; Vischer et al., 2022), MAST maintains a sparse network consistently throughout the entire training regimen. Consequently, MAST endows itself with a unique advantage, manifesting in a remarkable acceleration of training FLOPs. We observed up to 20-fold acceleration in training and inference FLOPs for MAST-QMIX in the `2s3z` task, with an average acceleration of 10-fold, 9-fold, and 8-fold for QMIX, WQMIX, and RES-QMIX, respectively. Moreover, MAST showcases significant model

compression ratios, achieving reductions in model size ranging from 5-fold to 20-fold for QMIX, WQMIX, and RES-QMIX, while incurring only minor performance trade-offs, typically below 3%.

## 5.2 ABLATION STUDY

We conduct a comprehensive ablation study on three critical elements of MAST: hybrid TD($\lambda$) targets, the Soft Mellowmax operator, and dual buffers, specifically evaluating their effects on QMIX and WQMIX. Notably, since MAST-QMIX shares similarities with MAST-RES, our experiments focus on QMIX and WQMIX within the `3s5z` task. This meticulous analysis seeks to elucidate the influence of each component on MAST and their robustness in the face of hyperparameter variations. The reported results are expressed as percentages and are normalized with respect to dense models.

**Hybrid TD($\lambda$)**  We commence our analysis by evaluating various burn-in times $T_0$, for hybrid TD($\lambda$). Additionally, we explore the impact of different $\lambda$ values within hybrid TD($\lambda$). The results are presented in Table 2, revealing hybrid TD($\lambda$) targets achieve optimal performance with a burn-in time of $T_0 = 0.75M$ and $\lambda = 0.6$. It is noteworthy that hybrid TD($\lambda$) targets lead to significant performance improvements in WQMIX, while their impact on QMIX is relatively modest.

Table 2: Ablation study on Hybrid TD($\lambda$).

| Alg. | $T_0$ | | | | $\lambda$ | | | | | |
|---|---|---|---|---|---|---|---|---|---|---|
| | 0 | 0.75M | 1.5M | 2M | 0 | 0.2 | 0.4 | 0.6 | 0.8 | 1 |
| QMIX / RES | 93.6 | **97.9** | 92.5 | 91.5 | 91.5 | 94.7 | 96.8 | 96.8 | **97.9** | 89.4 |
| WQMIX | 83.5 | **98.0** | 76.9 | 70.3 | 83.5 | 83.5 | 74.7 | **98.0** | 96.1 | 87.9 |
| Avg. | 88.5 | **97.9** | 84.7 | 80.9 | 87.5 | 89.1 | 85.7 | **97.4** | 97.0 | 88.6 |

**Soft Mellowmax Operator**  The Soft Mellowmax operator in Eq.(3) introduces two hyperparameters, $\alpha$ and $\omega$. A comprehensive examination of various parameter configurations is presented in Table3. Our analysis reveals that the performance of MAST exhibits robustness to changes in the two hyperparameters associated with the Soft Mellowmax operator.

Additionally, it is worth noting that the Softmax operator is also employed in (Pan et al., 2021) to mitigate overestimation in multi-agent Q learning. To examine the effectiveness of various operators, including max, Softmax, Mellowmax, and Soft Mellowmax, we conduct a comparative analysis in Figure 8. Our findings indicate that the Soft Mellowmax operator surpasses all other baselines in alleviating overestimation. Although the Softmax operator demonstrates similar performance to the Soft Mellowmax operator, it is important to note that the Softmax operator entails higher computational costs, as elucidated in Appendix B.4.5.

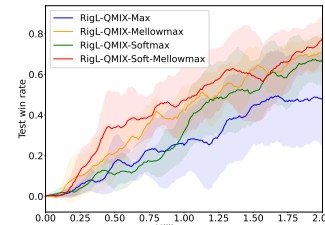

Figure 8: Comparison of different operators.

**Dual buffers**  It is worth noting that in each training step, we concurrently sample two batches from the two buffers, $\mathcal{B}_1$ and $\mathcal{B}_2$. We maintain a fixed total batch size of 32 while varying the sample partitions $b_1 : b_2$ within MAST. The results, detailed in Table 3, reveal that employing two buffers with a partition ratio of $5:3$ yields the best performance. Additionally, we observed a significant degradation in MAST's performance when using data solely from a single buffer, whether it be the online or offline buffer. This underscores the vital role of dual buffers in sparse MARL.

Table 3: Ablation study on dual buffers and Soft Mellowmax Operator.

| Alg. | Smaple Partitions | | | | Soft Mellowmax Operator | | | | |
|---|---|---|---|---|---|---|---|---|---|
| | $8:0$ | $5:3$ | $3:5$ | $0:8$ | $\alpha=1$ $\omega=10$ | $\alpha=5$ $\omega=5$ | $\alpha=5$ $\omega=10$ | $\alpha=10$ $\omega=5$ | $\alpha=10$ $\omega=10$ |
| QMIX / RES | 93.6 | **97.9** | 97.8 | 85.1 | 97.9 | **100.0** | 98.9 | 96.8 | 97.9 |
| WQMIX | 64.8 | **98.0** | 86.8 | 70.3 | **98.0** | 92.3 | 87.9 | 92.3 | 85.7 |
| Avg. | 79.2 | **97.9** | 92.3 | 77.7 | **97.9** | 96.1 | 93.4 | 94.5 | 91.8 |

## 5.3 SPARSE MODELS OBTAINED BY MAST

We conduct a comparative analysis of diverse sparse network architectures. With identical sparsity levels, distinct sparse architectures lead to different hypothesis spaces. As emphasized in

(Frankle & Carbin, 2019), specific architectures, such as the "winning ticket," outperform randomly generated counterparts. We compare three architectures: the "random ticket" (randomly sampled topology held constant during training), the "winning ticket" (topology from a MAST or RigL run and kept unchanged during training), and the "cheating ticket" (trained with MAST).

Figure 9 illustrates that both the "cheating ticket" and "winning ticket" by MAST achieve the highest performance, closely approaching the original dense model's performance. Importantly, using a fixed random topology during training fails to fully exploit the benefits of high sparsity, resulting in significant performance degradation. Furthermore, RigL's "winning ticket" fares poorly, akin to the "random ticket." These results underscore the advantages of our MAST approach, which automatically discovers effective sparse architectures through gradient-based topology evolution, without the need for pretraining methods like knowledge distillation (Schmitt et al., 2018).

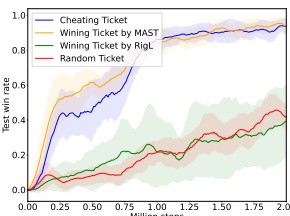

Figure 9: Comparison of different sparse masks.

Crucially, our MAST method incorporates key elements: the hybrid TD($\lambda$) mechanism, Soft Mellowmax operator, and dual buffers. Compared to RigL, these components significantly improve value estimation and training stability in sparse models facilitating efficient topology evolution.

Figure 10 showcases the evolving sparse mask of a hidden layer during MAST-QMIX training in 3s5z, capturing snapshots at 0, 5, 10, and 20 million steps. For additional layers, refer to Appendix B.8. The upper section of Figure 10 illustrates the mask, while the lower part presents connection counts for output dimensions, sorted in descending order. Notably, a pronounced shift in the mask is evident at the start of training, followed by a gradual convergence of connections within the layer onto a subset of input neurons. This convergence is discernible from the clustering of light pixels forming continuous rows in the lower segment of the final mask visualization, where several output dimensions exhibit minimal or no connections. This observation underscores the distinct roles played by various neurons in the representation process, emphasizing the prevalent redundancy in dense models and highlighting the effectiveness of our MAST framework.

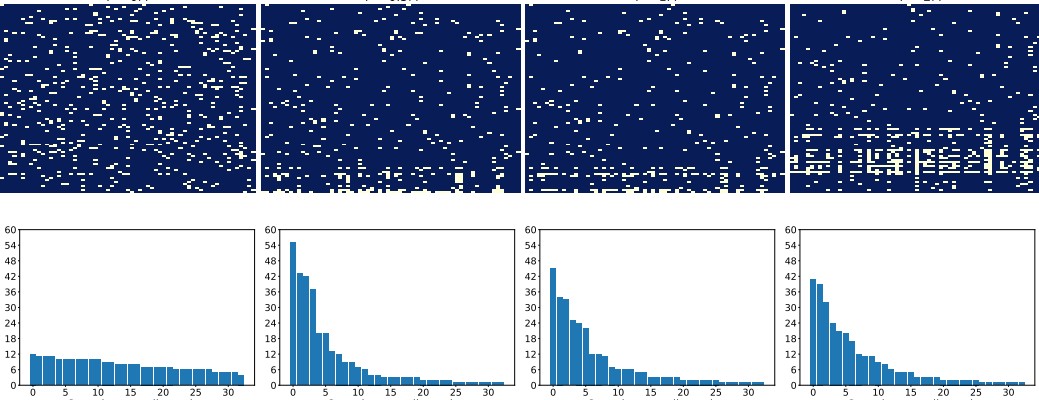

Figure 10: Part of first hidden layer weight masks in MAST-QMIX for agent 1. Upper part: Light pixels in row $i$ and column $j$ indicate the existence of the connection for input dimension $j$ and output dimension $i$, while the dark pixel represents the empty connection; Lower part: Number of nonzero connections for output dimensions in descending order.

## 6  CONCLUSION

This paper introduces MAST, a novel sparse training framework for deep MARL, utilizing gradient-based topology evolution to efficiently explore network configurations in sparse models. MARL faces significant challenges in ultra-sparse models, including value estimation errors and training instability. To address these, MAST offers innovative solutions: a hybrid TD($\lambda$) target mechanism combined with the Soft Mellowmax operator for precise value estimation in extreme sparsity, and a dual buffer mechanism for enhanced training stability. MAST enables efficient MARL agent training with minimal performance impact, employing ultra-sparse networks throughout. Our experiments across popular MARL algorithms validate MAST's leadership in sparse training, achieving model compression of $5\times$ to $20\times$ with minimal performance degradation, and up to a remarkable $20\times$ reduction in FLOPs for both training and inference. Besides, the limitation and future work of the MAST framework are discussed in Appendix A.4.

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

# Supplementary Materials

## A  ADDITIONAL DETAILS FOR MAST FRAMEWORK

### A.1  COMPREHENSIVE RELATED WORK

Sparse networks, initially proposed in deep supervised learning can train a 90%-sparse network without performance degradation from scratch. However, for deep reinforcement learning, the learning target is not fixed but evolves in a bootstrap way (Tesauro et al., 1995), and the distribution of the training data can also be non-stationary (Desai et al., 2019), which makes the sparse training more difficult. In the following, we list some representative works for training sparse models from supervised learning to reinforcement learning.

**Sparse Models in Supervised Learning**  Various techniques have been explored for creating sparse networks, ranging from pruning pre-trained dense networks (Han et al., 2015; 2016; Srinivas et al., 2017), to employing methods like derivatives (Dong et al., 2017; Molchanov et al., 2019), regularization (Louizos et al., 2018), dropout (Molchanov et al., 2017), and weight reparameterization (Schwarz et al., 2021). Another avenue of research revolves around the Lottery Ticket Hypothesis (LTH) (Frankle & Carbin, 2019), which posits the feasibility of training sparse networks from scratch, provided a sparse "winning ticket" initialization is identified. This hypothesis has garnered support in various deep learning models (Chen et al., 2020a; Brix et al., 2020). Additionally, there is a body of work dedicated to training sparse neural networks from the outset, involving techniques that evolve the structures of sparse networks during training. Examples include Deep Rewiring (DeepR) (Bellec et al., 2017), Sparse Evolutionary Training (SET) (Mocanu et al., 2018), Dynamic Sparse Reparameterization (DSR) (Mostafa & Wang, 2019), Sparse Networks from Scratch (SNFS) (Dettmers & Zettlemoyer, 2019), and Rigged Lottery (RigL) (Evci et al., 2020). Furthermore, methods like Single-Shot Network Pruning (SNIP) (Lee et al., 2019) and Gradient Signal Preservation (GraSP) (Wang et al., 2020) are geared towards identifying static sparse networks prior to training.

**Sparse Models in Single-Agent RL**  Existing research (Schmitt et al., 2018; Zhang et al., 2019) has employed knowledge distillation with static data to ensure training stability and generate small dense agents. Policy Pruning and Shrinking (PoPs) (Livne & Cohen, 2020) generates sparse agents through iterative policy pruning, while the LTH in DRL is first indentified in (Yu et al., 2020).

Another line of investigation aims to train sparse DRL models from scratch, eliminating the necessity of pre-training a dense teacher. Specifically, (Sokar et al., 2022) introduces the Sparse Evolutionary Training (SET) approach, achieving a remarkable $50\%$ sparsity level through topology evolution in DRL. Additionally, (Graesser et al., 2022) observes that pruning often yields superior results, with plain dynamic sparse training methods, including SET and RigL, significantly outperforming static sparse training approaches. More recently, RLx2 (Tan et al., 2022) has demonstrated the capacity to train DRL agents with highly sparse neural networks from scratch. Nevertheless, the application of RLx2 in MARL yields poor results, as demonstrated in Section 5.1.

**Sparse Models in MARL**   Existing works have made attempts to train sparse MARL agents, such as (Yang et al., 2022), which prunes networks for multiple agents during training, employing weight grouping (Wang et al., 2019). Another avenue of sparse MARL research seeks to enhance the scalability of MARL algorithms through sparse architectural modifications. For instance, (Sun et al., 2020) proposes the use of a sparse communication graph with graph neural networks to reduce problem scale. (Kim & Sung, 2023) adopts structured pruning for a deep neural network to extend the scalability. Yet another strand of sparse MARL focuses on parameter sharing between agents to reduce the number of trainable parameters, with representative works including (Gupta et al., 2017; Li et al., 2021; Christianos et al., 2021). However, existing methods fail to maintain high sparsity throughout the training process, such that the FLOPs reduction during training is incomparable to the MAST framework outlined in our paper.

## A.2 DECENTRALIZED PARTIALLY OBSERVABLE MARKOV DECISION PROCESS

We model the MARL problem as a decentralized partially observable Markov decision process (Dec-POMDP) (Oliehoek et al., 2016), represented by a tuple $\langle \mathcal{N}, \mathcal{S}, \mathcal{U}, P, r, \mathcal{Z}, O, \gamma \rangle$, where $\mathcal{N} = \{1, \ldots, N\}$ denotes the finite set of agents, $\mathcal{S}$ is the global state space, $\mathcal{U}$ is the action space for an agent, $P$ is the transition probability, $r$ is the reward function, $\mathcal{Z}$ is the observation space for an agent, $O$ is the observation function, and and $\gamma \in [0, 1)$ is the discount factor. At each timestep $t$, each agent $i \in \mathcal{N}$ receives an observation $z \in \mathcal{Z}$ from the observation function $O(s, i) : \mathcal{S} \times \mathcal{N} \mapsto \mathcal{Z}$ due to partial observability, and chooses an action $u_i \in \mathcal{U}$, which forms a joint action $\boldsymbol{u} \in \mathcal{U} \equiv \mathcal{U}^n$. The joint action $\boldsymbol{u}$ taken by all agents leads to a transition to the next state $s'$ according to transition probability $P(s' \mid s, \boldsymbol{u}) : \mathcal{S} \times \mathcal{U} \times \mathcal{S} \mapsto [0, 1]$ and a joint reward $r(s, \boldsymbol{u}) : \mathcal{S} \times \mathcal{U} \mapsto \mathbb{R}$. As the time goes by, each agent $i \in \mathcal{N}$ has an action-observation history $\tau_i \in \mathcal{T} \equiv (\mathcal{Z} \times \mathcal{U})^*$, where $\mathcal{T}$ is the history space. Based on $\tau_i$, each agent $i$ outputs an action $u_i$ according to its constructed policy $\pi_i(u_i \mid \tau_i) : \mathcal{T} \times \mathcal{U} \mapsto [0, 1]$. The goal of agents is to find an optimal joint policy $\boldsymbol{\pi} = \langle \pi_1, \ldots, \pi_N \rangle$, which maximize the joint cumulative rewards $J(s_0; \boldsymbol{\pi}) = \mathbb{E}_{\boldsymbol{u}_t \sim \boldsymbol{\pi}(\cdot|s_t), s_{t+1} \sim P(\cdot|s_t, \boldsymbol{u}_t)} \left[ \sum_{t=0}^{\infty} \gamma^i r(s_t, \boldsymbol{u}_t) \right]$, where $s_0$ is the initial state. The joint action-value function associated with policy $\boldsymbol{\pi}$ is defined as $Q^{\boldsymbol{\pi}}(s_t, \boldsymbol{u}_t) = \mathbb{E}_{\boldsymbol{u}_{t+i} \sim \boldsymbol{\pi}(\cdot|s_{t+i}), s_{t+i+1} \sim P(\cdot|s_{t+i}, \boldsymbol{u}_{t+i})} \left[ \sum_{i=0}^{\infty} \gamma^i r(s_{t+i}, \boldsymbol{u}_{t+i}) \right]$.

## A.3 MAST WITH DIFFERENT ALGORITHMS

In this section, we present the pseudocode implementations of MAST for QMIX (Rashid et al., 2020b) and WQMIX (Rashid et al., 2020a) in Algorithm 2 and Algorithm 3, respectively. It is noteworthy that RES (Pan et al., 2021) exclusively modifies the training target without any alterations to the learning protocol or network structure. Consequently, the implementation of MAST with RES mirrors that of QMIX.

Crucially, MAST stands as a versatile sparse training framework, applicable to a range of value decomposition-based MARL algorithms, extending well beyond QMIX, WQMIX[1], and RES. Furthermore, MAST's three innovative components—hybrid TD($\lambda$), Soft Mellowmax operator, and dual buffer—can be employed independently, depending on the specific algorithm's requirements. This flexible framework empowers the training of sparse networks from the ground up, accommodating a wide array of MARL algorithms.

In the following, we delineate the essential steps of implementing MAST with QMIX (Algorithm 2). The steps for WQMIX are nearly identical, with the exception of unrestricted agent networks and

---

[1]Note that WQMIX encompasses two distinct instantiations, namely Optimistically-Weighted (OW) QMIX and Centrally-Weighted (CW) QMIX. In this paper, we specifically focus on OWQMIX.

the unrestricted mixing network's inclusion. Also, note that we follow the symbol definitions from (Colom, 2021) in Algorithm 2 and 3.

**Gradient-based Topology Evolution:** The process of topology evolution is executed within Lines 31-33 in Algorithm 2. Specifically, the topology evolution update occurs at intervals of $\Delta_m$ timesteps. For a comprehensive understanding of additional hyperparameters pertaining to topology evolution, please refer to the definitions provided in Algorithm 1.

---

**Algorithm 2** MAST-QMIX

---

1: Initialize sparse agent networks, mixing network and hypernetwork with random parameters $\theta$ and random masks $M_\theta$ with determined sparsity $S$ .
2: $\hat{\theta} \leftarrow \theta \odot M_\theta$ *// Start with a random sparse network*
3: Initialize target networks $\hat{\theta}^- \leftarrow \hat{\theta}$
4: Set the learning rate to $\alpha$
5: Initialize the replay buffer $\mathcal{B}_1 \leftarrow \{\}$ with large capacity $C_1$ and $\mathcal{B}_2 \leftarrow \{\}$ with small capacity $C_2$
6: Initialize training step $\leftarrow 0$
7: **while** step $< T_{max}$ **do**
8:     $t \leftarrow 0$
9:     $s_0 \leftarrow$ initial state
10:     **while** $s_t \neq$ terminal and $t<$ episode limit **do**
11:       **for** each agent a **do**
12:         $\tau_t^a \leftarrow \tau_{t-1}^a \cup \{(o_t, u_{t-1})\}$
13:         $\epsilon \leftarrow$ epsilon-schedule(step)
14:         $u_t^a \leftarrow \begin{cases} \mathrm{argmax}_{u_t^a} Q\left(\tau_t^a, u_t^a\right) & \text{with probability } 1 - \epsilon \\ \mathrm{randint}(1, |U|) & \text{with probability } \epsilon \end{cases}$   *// $\epsilon$-greedy exploration*
15:       **end for**
16:     Get reward $r_t$ and next state $s_{t+1}$
17:     $\mathcal{B}_1 \leftarrow \mathcal{B}_1 \cup \{(s_t, \mathbf{u}_t, r_t, s_{t+1})\}$ *// Data in the buffer is of episodes form.*
18:     $\mathcal{B}_2 \leftarrow \mathcal{B}_2 \cup \{(s_t, \mathbf{u}_t, r_t, s_{t+1})\}$
19:     $t \leftarrow t + 1$,step $\leftarrow$ step $+ 1$
20:     **end while**
21:     **if** $|\mathcal{B}_1| >$ batch-size **then**
22:       $b \leftarrow$ random batch of episodes from $\mathcal{B}_1$ and $\mathcal{B}_2$ *// Sample from dual buffers.*
23:       **for** each timestep $t$ in each episode in batch $b$ **do**
24:

$$Q_{tot} \leftarrow \text{Mixing-network}\left((Q_1(\tau_t^1, u_t^1), \cdots, Q_n(\tau_t^n, u_t^n)); \text{Hypernetwork}(s_t; \hat{\theta})\right)$$

25:         Compute TD target $y$ according to Eq. (6). *// TD($\lambda$) targets with Soft Mellowmax operator.*
26:       **end for**
27:       $\Delta Q_{tot} \leftarrow y - Q_{tot}$
28:       $\Delta\hat{\theta} \leftarrow \nabla_{\hat{\theta}} \frac{1}{b} \sum (\Delta Q_{tot})^2$
29:       $\hat{\theta} \leftarrow \hat{\theta} - \alpha \Delta\hat{\theta}$
30:     **end if**
31:     **if** step $\mod \Delta_m = 0$ **then**
32:       Topology_Evolution(networks$_{\hat{\theta}}$ by Algorithm 1.
33:     **end if**
34:     **if** step $\mod I = 0$, where is the target network update interval **then**
35:       $\hat{\theta}^- \leftarrow \hat{\theta}$ *// Update target network.*
36:       $\hat{\theta}^- \leftarrow \hat{\theta}^- \odot M_{\hat{\theta}}$
37:     **end if**
38: **end while**

---

---

**Algorithm 3** MAST-(OW)QMIX

---

1: Initialize sparse agent networks, mixing network and hypernetwork with random parameters $\theta$ and random masks $M_\theta$ with determined sparsity $S$.
2: Initialize unrestricted agent networks and unrestricted mixing network with random parameters $\phi$ and random masks $M_\phi$ with determined sparsity $S$.
3: $\hat{\theta} \leftarrow \theta \odot M_\theta$ , $\hat{\phi} \leftarrow \phi \odot M_\phi$ *// Start with a random sparse network*
4: Initialize target networks $\hat{\theta}^- \leftarrow \hat{\theta}$, $\hat{\phi}^- \leftarrow \hat{\phi}$
5: Set the learning to rate $\alpha$
6: Initialize the replay buffer $\mathcal{B}_1 \leftarrow \{\}$ with large capacity $C_1$ and $\mathcal{B}_2 \leftarrow \{\}$ with small capacity $C_2$
7: Initialize training step $\leftarrow 0$
8: **while** step $< T_{max}$ **do**
9: $\quad t \leftarrow 0$,
10: $\quad s_0 \leftarrow$ initial state
11: $\quad$ **while** $s_t \neq$ terminal and $t<$ episode limit **do**
12: $\quad\quad$ **for** each agent a **do**
13: $\quad\quad\quad \tau_t^a \leftarrow \tau_{t-1}^a \cup \{(o_t, u_{t-1})\}$
14: $\quad\quad\quad \epsilon \leftarrow$ epsilon-schedule(step)
15: $\quad\quad\quad u_t^a \leftarrow \begin{cases} \operatorname{argmax}_{u_t^a} Q(\tau_t^a, u_t^a; \hat{\theta}) & \text{with probability } 1 - \epsilon \\ \operatorname{randint}(1, |U|) & \text{with probability } \epsilon \end{cases}$ *// $\epsilon$-greedy exploration*
16: $\quad\quad$ **end for**
17: $\quad\quad$ Get reward $r_t$ and next state $s_{t+1}$
18: $\quad\quad \mathcal{B}_1 \leftarrow \mathcal{B}_1 \cup \{(s_t, \mathbf{u}_t, r_t, s_{t+1})\}$ *// Data in the buffer is of episodes form.*
19: $\quad\quad \mathcal{B}_2 \leftarrow \mathcal{B}_2 \cup \{(s_t, \mathbf{u}_t, r_t, s_{t+1})\}$
20: $\quad\quad t \leftarrow t + 1$, step $\leftarrow$ step $+ 1$
21: $\quad$ **end while**
22: $\quad$ **if** $|\mathcal{B}_1| >$ batch-size **then**
23: $\quad\quad b \leftarrow$ random batch of episodes from $\mathcal{B}_1$ and $\mathcal{B}_2$ *// Sample from dual buffers.*
24: $\quad\quad$ **for** each timestep $t$ in each episode in batch $b$ **do**
25:
$$Q_{tot} \leftarrow \text{Mixing-network}\left((Q_1(\tau_t^1, u_t^1; \hat{\theta}), ..., Q_n(\tau_t^n, u_t^n; \hat{\theta})); \text{Hypernetwork}(s_t; \hat{\theta})\right)$$

26:
$$\hat{Q}^* \leftarrow \text{Unrestricted-Mixing-network}\left(Q_1(\tau_t^1, u_t^1; \hat{\phi}), ..., Q_n(\tau_t^n, u_t^n; \hat{\phi}), s_t\right)$$

27: $\quad\quad\quad$ Compute TD target $y$ with target Unrestricted-Mixing network according to Eq. (6). *// TD($\lambda$) targets with Soft Mellowmax operator.*
28: $\quad\quad\quad \omega(s_t, \mathbf{u_t}) \leftarrow \begin{cases} 1, & Q_{tot} < y \\ \alpha, & \text{otherwise.} \end{cases}$
29: $\quad\quad$ **end for**
30: $\quad\quad \Delta Q_{tot} \leftarrow y - Q_{tot}$
31: $\quad\quad \Delta\hat{\theta} \leftarrow \nabla_{\hat{\theta}} \frac{1}{b} \sum \omega(s, \mathbf{u})(\Delta Q_{tot})^2$
32: $\quad\quad \hat{\theta} \leftarrow \hat{\theta} - \alpha\Delta\hat{\theta}$
33: $\quad\quad \Delta\hat{Q}^* \leftarrow y - \hat{Q}^*$
34: $\quad\quad \Delta\hat{\phi} \leftarrow \nabla_{\hat{\phi}} \frac{1}{b} \sum (\Delta\hat{Q}^*)^2$
35: $\quad\quad \hat{\phi} \leftarrow \hat{\phi} - \alpha\Delta\hat{\phi}$
36: $\quad$ **end if**
37: $\quad$ **if** step $\mod \Delta_m = 0$ **then**
38: $\quad\quad$ Topology_Evolution(networks$_{\hat{\theta}}$) and Topology_Evolution(networks$_{\hat{\phi}}$) by Algorithm 1.
39: $\quad$ **end if**
40: $\quad$ **if** step $\mod I = 0$, where is the target network update interval **then**
41: $\quad\quad \hat{\theta}^- \leftarrow \hat{\theta}, \hat{\phi}^- \leftarrow \hat{\phi}$
42: $\quad\quad \hat{\theta}^- \leftarrow \hat{\theta}^- \odot M_{\hat{\theta}}, \hat{\phi}^- \leftarrow \hat{\phi}^- \odot M_{\hat{\phi}}$
43: $\quad$ **end if**
44: **end while**

---

**TD Targets:** Hybrid TD($\lambda$) with Soft Mellowmax operator is computed in the Line 25 in Algorithm 2, which modify the TD target $y$ as follows:

$$y_{\mathrm{S}} = \begin{cases} G_t^{(1)}, & \text{if } t < T_0. \\ (1-\lambda) \sum_{n=1}^{\infty} \lambda^{n-1} \mathcal{T}_t^{(n)}, & \text{Otherwise.} \end{cases} \tag{6}$$

Here, $\lambda \in [0, 1]$ is a hyperparameter, and

$$\mathcal{T}_t^{(n)} = \sum_{i=t}^{t+n} \gamma^{i-t} r_i + \gamma^{n+1} f_s \left( \mathrm{sm}_\omega(\bar{Q}_1(\tau_1, \cdot)), \dots, \mathrm{sm}_\omega(\bar{Q}_N(\tau_N, \cdot)) \right), \tag{7}$$

where $f_s$ denotes the mixing network and $\bar{Q}_i$ is the target network of $Q_i$. The loss function of MAST, $\mathcal{L}_{\mathrm{S}}(\theta)$, is defined as:

$$\mathcal{L}_{\mathrm{S}}(\theta) = \mathbb{E}_{(s,\boldsymbol{u},r,s') \sim \mathcal{B}_1 \cup \mathcal{B}_2} \left[ (y_{\mathrm{S}} - Q_{tot}(s, \boldsymbol{u}))^2 \right] \tag{8}$$

**Dual Buffers:** With the creation of two buffers $\mathcal{B}_1$ and $\mathcal{B}_2$, the gradient update with data sampled from dual buffers is performed in Lines 21-30 in Algorithm 2.

### A.4 LIMITATIONS OF MAST

This paper introduces MAST, a novel framework for sparse training in deep MARL, leveraging gradient-based topology evolution to explore network configurations efficiently. However, understanding its limitations is crucial for guiding future research efforts.

**Hyperparameters:** MAST relies on multiple hyperparameters for its key components: topology evolution, TD($\lambda$) targets with Soft Mellowmax Operator, and dual buffers. Future work could explore methods to automatically determine these hyperparameters or streamline the sparse training process with fewer tunable settings.

**Implementation:** While MAST achieves efficient MARL agent training with minimal performance trade-offs using ultra-sparse networks surpassing 90% sparsity, its current use of unstructured sparsity poses challenges for running acceleration. The theoretical reduction in FLOPs might not directly translate to reduced running time. Future research should aim to implement MAST in a structured sparsity pattern to bridge this gap between theoretical efficiency and practical implementation.

## B EXPERIMENTAL DETAILS

In this section, we offer comprehensive experimental insights, encompassing hardware configurations, environment specifications, hyperparameter settings, model size computations, FLOPs calculations, and supplementary experimental findings.

### B.1 HARDWARE SETUP

Our experiments are implemented with PyTorch 2.0.0 (Paszke et al., 2017) and run on $4\times$ NVIDIA GTX Titan X (Pascal) GPUs. Each run needs about $12 \sim 24$ hours for QMIX or WQMIX, and about $24 \sim 72$ hours for RES for two million steps. depends on the environment types. The code will be open-sourced upon publication of the paper.

### B.2 ENVIRONMENT

We assess the performance of our MAST framework using the SMAC benchmark (Samvelyan et al., 2019), a dedicated platform for collaborative multi-agent reinforcement learning research based on Blizzard's StarCraft II real-time strategy game, specifically version 4.10. It is important to note that performance may vary across different versions. Our experimental evaluation encompasses four distinct maps, each of which is described in detail below.

- `3m`: An easy map, where the agents are 3 Marines, and the enemiesa are 3 Marines.
- `2s3z`: An easy map, where the agents are 2 Stalkers and 3 Zealots, and the enemies are 2 Stalkers and 3 Zealots.
- `3s5z`: An easy map, where the agents are 3 Stalkers and 5 Zealots, and the enemies are 3 Stalkers and 5 Zealots.
- `2c_vs_64zg`: A hard map, where the agents are 2 Colossi, and the enemies are 64 Zerglings.

### B.3 HYPERPARAMETER SETTINGS

Table 5 provides a comprehensive overview of the hyperparameters employed in our experiments for MAST-QMIX, MAST-WQMIX, and MAST-RES. It includes detailed specifications for network parameters, RL parameters, and topology evolution parameters, allowing for a thorough understanding of our configurations. Besides, MAST is implemented based on the PyMARL (Samvelyan et al., 2019) framework with the same network structures and hyperparameters as given in Table 5. We also provide a hyperparameter recommendation for three key components, i.e. gradient-based topology evolution, Soft Mellowmax enabled hybrid TD($\lambda$) targets and dual buffers, in Table 4 for deployment MAST framework in other problems.

Table 4: Recommendation for Key Hyperparameters in MAST.

| Category | Hyperparameter | Value |
|---|---|---|
| Topology Evolution | Initial mask update fraction $\zeta_0$ | 0.5 |
| | Mask update interval $\Delta_m$ | 200 episodes |
| TD Targets | Burn-in time $T_0$ | 3/8 of total training steps |
| | $\lambda$ value in TD($\lambda$) | 0.6 or 0.8 |
| | $\alpha$ in soft mellow-max operator | 1 |
| | $\omega$ in soft mellow-max operator | 10 |
| Dual Buffer | Offline buffer size $C_1$ | $5 \times 10^3$ episodes |
| | Online buffer size $C_2$ | 128 episodes |
| | Sample partition of online and offline buffer | 3:5 |

### B.4 CALCULATION OF MODEL SIZES AND FLOPS

#### B.4.1 MODEL SIZE

First, we delineate the calculation of model sizes, which refers to the total number of parameters within the model.

- For a sparse network with $L$ fully-connected layers, the model size, as expressed in prior works (Evci et al., 2020; Tan et al., 2022), can be computed using the equation:

$$M_{\text{linear}} = \sum_{l=1}^{L} (1 - S_l) I_l O_l, \tag{9}$$

  where $S_l$ represents the sparsity, $I_l$ is the input dimensionality, and $O_l$ is the output dimensionality of the $l$-th layer.

- For a sparse network with $L$ GRU layers, considering the presence of 3 gates in a single layer, the model size can be determined using the equation:

$$M_{\text{GRU}} = \sum_{l=1}^{L} (1 - S_l) \times 3 \times h_l \times (h_l + I_l), \tag{10}$$

  where $h_l$ represents the hidden state dimensionality.

Specifically, the "Total Size" column in Table 1 within the manuscript encompasses the model size, including both agent and mixing networks during training. For QMIX, WQMIX, and RES, target

Table 5: Hyperparameters of MAST-QMIX, MAST-WQMIX and MAST-RES.

| Category | Hyperparameter | Value |
|---|---|---|
| Shared Hyperparameters | Optimizer | RMSProp |
| | Learning rate $\alpha$ | $5 \times 10^{-4}$ |
| | Discount factor $\gamma$ | 0.99 |
| | Number of hidden units per layer of agent network | 64 |
| | Hidden dimensions in the GRU layer of agent network | 64 |
| | Embedded dimensions of mixing network | 32 |
| | Hypernet layers of mixing network | 2 |
| | Embedded dimensions of hypernetwork | 64 |
| | Activation Function | ReLU |
| | Batch size $B$ | 32 episodes |
| | Warmup steps | 50000 |
| | Initial $\epsilon$ | 1.0 |
| | Final $\epsilon$ | 0.05 |
| | Double DQN update | True |
| | Target network update interval $I$ | 200 episodes |
| | Initial mask update fraction $\zeta_0$ | 0.5 |
| | Mask update interval $\Delta_m$ | timesteps of 200 episodes |
| | Offline buffer size $C_1$ | $5 \times 10^3$ episodes |
| | Online buffer size $C_2$ | 128 episodes |
| | Burn-in time $T_0$ | $7.5 \times 10^5$ |
| | $\alpha$ in soft mellow-max operator | 1 |
| | $\omega$ in soft mellow-max operator | 10 |
| | Number of episodes in a sampled batch of offline buffer $S_1$ | 20 |
| | Number of episodes in a sampled batch of online buffer $S_2$ | 12 |
| Hyperparameters for MAST-QMIX | Linearly annealing steps for $\epsilon$ | 50k |
| | $\lambda$ value in TD($\lambda$) | 0.8 |
| Hyperparameters for MAST-WQMIX | Linearly annealing steps for $\epsilon$ | 100k |
| | $\lambda$ value in TD($\lambda$) | 0.6 |
| | Coefficient of $Q_{tot}$ loss | 1 |
| | Coefficient of $\hat{Q}^*$ loss | 1 |
| | Embedded dimensions of unrestricted mixing network | 256 |
| | Embedded number of actions of unrestricted agent network | 1 |
| | $\alpha$ in weighting function | 0.1 |
| Hyperparameters for MAST-RES | Linearly annealing steps for $\epsilon$ | 50k |
| | $\lambda$ value in TD($\lambda$) | 0.8 |
| | $\lambda$ value in Softmax operator | 0.05 |
| | Inverse temperature $\beta$ | 5.0 |

networks are employed as target agent networks and target mixing networks. We denote the model sizes of the agent network, mixing network, unrestricted agent network, and unrestricted mixing network as $M_{\text{Agent}}$, $M_{\text{Mix}}$, $M_{\text{Unrestricted-Agent}}$, and $M_{\text{Unrestricted-Mix}}$, respectively. Detailed calculations of these model sizes are provided in the second column of Table 6.

Table 6: FLOPs and model size for MAST-QMIX , MAST-WQMIX and MAST-RES.

| Algorithm | Model size | Training FLOPs | Inference FLOPs |
|---|---|---|---|
| MAST-QMIX | $2M_{\text{Agent}} + 2M_{\text{Mix}}$ | $4B(\text{FLOPs}_{\text{Agent}} + \text{FLOPs}_{\text{Mix}})$ | $\text{FLOPs}_{\text{Agent}}$ |
| MAST-WQMIX | $M_{\text{Agent}} + M_{\text{Mix}}+$ $2M_{\text{Unrestricted-Agent}}+$ $2M_{\text{Unrestricted-Mix}}$ | $3B(\text{FLOPs}_{\text{Agent}} + \text{FLOPs}_{\text{Mix}})+$ $4B \cdot \text{FLOPs}_{\text{Unrestricted-Agent}}+$ $4B \cdot \text{FLOPs}_{\text{Unrestricted-Mix}}$ | $\text{FLOPs}_{\text{Agent}}$ |
| MAST-RES | $2M_{\text{Agent}} + 2M_{\text{Mix}}$ | $4B \cdot \text{FLOPs}_{\text{Agent}}+$ $(5 + nm)B \cdot \text{FLOPs}_{\text{Mix}}$ | $\text{FLOPs}_{\text{Agent}}$ |

### B.4.2 FLOPs Calculation

Initially, for a sparse network with $L$ fully-connected layers, the required FLOPs for a forward pass are computed as follows (also adopted in (Evci et al., 2020) and (Tan et al., 2022)):

$$\text{FLOPs} = \sum_{l=1}^{L}(1 - S_l)(2I_l - 1)O_l, \tag{11}$$

where $S_l$ is the sparsity, $I_l$ is the input dimensionality, and $O_l$ is the output dimensionality of the $l$-th layer. Similarly, for a sparse network with $L$ GRU (Chung et al., 2014) layers, considering the presence of 3 gates in a single layer, the required FLOPs for a forward pass are:

$$\text{FLOPs} = \sum_{l=1}^{L}(1 - S_l) \times 3 \times h_l \times [2(h_l + I_l) - 1], \tag{12}$$

where $h_l$ is the hidden state dimensionality.

We denote $B$ as the batch size employed in the training process, and $\text{FLOPs}_{\text{Agent}}$ and $\text{FLOPs}_{\text{Mix}}$ as the FLOPs required for a forward pass in the agent and mixing networks, respectively. The inference FLOPs correspond exactly to $\text{FLOPs}_{\text{Agent}}$, as detailed in the last column of Table 6. When it comes to training FLOPs, the calculation encompasses multiple forward and backward passes across various networks, which will be thoroughly elucidated later. Specifically, we compute the FLOPs necessary for each training iteration. Additionally, we omit the FLOPs associated with the following processes, as they exert minimal influence on the ultimate result:

- **Interaction with the environment:** This operation, where agents decide actions for interaction with the environment, incurs FLOPs equivalent to $\text{FLOPs}_{\text{Agent}}$. Notably, this value is considerably smaller than the FLOPs required for network updates, as evident in Table 6, given that $B \gg 1$.

- **Updating target networks:** Each parameter in the networks is updated as $\theta' \leftarrow \theta$. Consequently, the number of FLOPs in this step mirrors the model size, and is thus negligible.

- **Topology evolution:** This element is executed every 200 gradient updates. To be precise, the average FLOPs involved in topology evolution are computed as $B \times \frac{2\text{FLOPs}_{\text{Agent}}}{(1-S^{(a)})\Delta_m}$ for the agent, and $B \times \frac{2\text{FLOPs}_{\text{Mix}}}{(1-S^{(m)})\Delta_m}$ for the mixer. Given that $\Delta_m = 200$, the FLOPs incurred by topology evolution are negligible.

Therefore, our primary focus shifts to the FLOPs related to updating the agent and mixer. We will first delve into the details for QMIX, with similar considerations for WQMIX and RES.

### B.4.3 Training FLOPs Calculation in QMIX

Recall the way to update networks in QMIX is given by

$$\theta \leftarrow \theta - \alpha \nabla_\theta \frac{1}{B}\sum(y_t - Q_{tot}(s_i, a_i; \theta))^2, \tag{13}$$

where $B$ is the batch size. Subsequently, we can compute the FLOPs of training as:

$$\text{FLOPs}_{\text{train}} = \text{FLOPs}_{\text{TD\_target}} + \text{FLOPs}_{\text{compute\_loss}} + \text{FLOPs}_{\text{backward\_pass}}, \tag{14}$$

where $\text{FLOPs}_{\text{TD\_target}}$, $\text{FLOPs}_{\text{compute\_loss}}$, and $\text{FLOPs}_{\text{backward\_pass}}$ refer to the numbers of FLOPs in computing the TD targets in forward pass, loss function in forward pass, and gradients in backward pass (backward-propagation), respectively. By Eq. (6) and (8), we have:

$$\begin{aligned} \text{FLOPs}_{\text{TD\_target}} &= B \times (\text{FLOPs}_{\text{Agent}} + \text{FLOPs}_{\text{Mix}}), \\ \text{FLOPs}_{\text{compute\_loss}} &= B \times (\text{FLOPs}_{\text{Agent}} + \text{FLOPs}_{\text{Mix}}). \end{aligned} \tag{15}$$

For the FLOPs of gradients backward propagation, $\text{FLOPs}_{\text{backward\_pass}}$, we compute it as two times the computational expense of the forward pass, which is adopted in existing literature (Evci et al., 2020), i.e.,

$$\text{FLOPs}_{\text{backward\_pass}} = B \times 2 \times (\text{FLOPs}_{\text{Agent}} + \text{FLOPs}_{\text{Mix}}), \tag{16}$$

Combining Eq. (14), Eq. (15), and Eq. (16), the FLOPs of training in QMIX is:

$$\text{FLOPs}_{\text{train}} = B \times 4 \times (\text{FLOPs}_{\text{Agent}} + \text{FLOPs}_{\text{Mix}}). \tag{17}$$

### B.4.4 TRAINING FLOPs CALCULATION IN WQMIX

The way to update the networks in WQMIX is different from that in QMIX. Specifically, denote the parameters of the original network and unrestricted network as $\theta$ and $\phi$, respectively, which are updated according to

$$\begin{aligned} \theta &\leftarrow \theta - \alpha \nabla_\theta \frac{1}{B} \sum_i \omega(s_i, a_i)(\mathcal{T}_t^\lambda - Q_{tot}(s_i, a_i; \theta))^2 \\ \phi &\leftarrow \phi - \alpha \nabla_\phi \frac{1}{B} \sum_i (\mathcal{T}_t^\lambda - \hat{Q}^*(s_i, a_i; \phi))^2 \end{aligned}, \tag{18}$$

where $B$ is the batch size, $\omega$ is the weighting function, $\hat{Q}^*$ is the unrestricted joint action value function. As shown in Algorithm 3, the way to compute TD target in WQMIX is different from that in QMIX. Thus, we have

$$\text{FLOPs}_{\text{TD\_target}} = B \times (\text{FLOPs}_{\text{Unrestricted-Agent}} + \text{FLOPs}_{\text{Unrestricted-Mix}}). \tag{19}$$

In this paper, we take an experiment on one of two instantiations of QMIX. i.e., OW-QMIX (Rashid et al., 2020a). Thus, the number of FLOPs in computing loss is

$$\text{FLOPs}_{\text{compute\_loss}} = B \times (\text{FLOPs}_{\text{Agent}} + \text{FLOPs}_{\text{Mix}} + \text{FLOPs}_{\text{Unrestricted-Agent}} + \text{FLOPs}_{\text{Unrestricted-Mix}}). \tag{20}$$

where unrestricted-agent and unrestricted-mix have similar network architectures as $Q_{tot}$ and $Q_{tot}$ to, respevtively. The FLOPs of gradients backward propagation can be given as

$$\text{FLOPs}_{\text{backward\_pass}} = B \times 2 \times (\text{FLOPs}_{\text{Agent}} + \text{FLOPs}_{\text{Mix}} + \text{FLOPs}_{\text{Unrestricted-Agent}} + \text{FLOPs}_{\text{Unrestricted-Mix}}). \tag{21}$$

Thus, the FLOPs of training in WQMIX can be computed by

$$\text{FLOPs}_{\text{train}} = B \times (3\text{FLOPs}_{\text{Agent}} + 3\text{FLOPs}_{\text{Mix}} + 4\text{FLOPs}_{\text{Unrestricted-Agent}} + 4\text{FLOPs}_{\text{Unrestricted-Mix}}). \tag{22}$$

### B.4.5 TRAINING FLOPs CALCULATION IN RES

Calculations of FLOPs for RES are similar to those in QMIX. The way to update the network parameter in RES is:

$$\theta \leftarrow \theta - \alpha \nabla_\theta \frac{1}{B} \sum_i (\mathcal{T}_t^\lambda - Q_{tot}(s_i, a_i; \theta))^2, \tag{23}$$

where $B$ is the batch size. Meanwhile, note that the way to compute TD target in RES (Pan et al., 2021) includes computing the *approximate Softmax operator*, we have:

$$\text{FLOPs}_{\text{TD\_target}} = B \times (\text{FLOPs}_{\text{Agent}} + n \times m \times (2\text{FLOPs}_{\text{Mix}})), \tag{24}$$

where $n$ is the number of agents, $m$ is the maximum number of actions an agent can take in a scenario. Other terms for updating networks are the same as QMIX. Thus, the FLOPs of training in RES can be computed by

$$\text{FLOPs}_{\text{train}} = B \times (4\text{FLOPs}_{\text{Agent}} + (3 + 2 \times n \times m)\text{FLOPs}_{\text{Mix}}). \tag{25}$$

B.5 TRAINING CURVES OF COMPARATIVE EVALUATION IN SECTION 5.1

Figure 11, Figure 12, and Figure 13 show the training curves of different algorithms in four SMAC environments. The performance is calculated as the average win rate per episode over the last 20 evaluations of the training. MAST outperforms baseline algorithms on all four environments in all three algorithms. We smooth the training curve by a 1-D filter by `scipy.signal.savgol_filter` in Python with `window_length=21` and `polyorder=2`.

These figures unequivocally illustrate MAST's substantial performance superiority over all baseline methods in all four environments across the three algorithms. Notably, static sparse (`SS`) consistently exhibit the lowest performance on average, highlighting the difficulty of finding optimal sparse network topologies in the context of sparse MARL models. Dynamic sparse training methods, namely `SET` and `RigL`, slightly outperform (`SS`), although their performance remains unsatisfactory. Sparse networks also, on average, underperform tiny dense networks. However, MAST significantly outpaces all other baselines, indicating the successful realization of accurate value estimation through our MAST method, which effectively guides gradient-based topology evolution. Notably, the single-agent method `RLx2` consistently delivers subpar results in all experiments, potentially due to its limited replay buffer capacity, severely hampering sample efficiency.

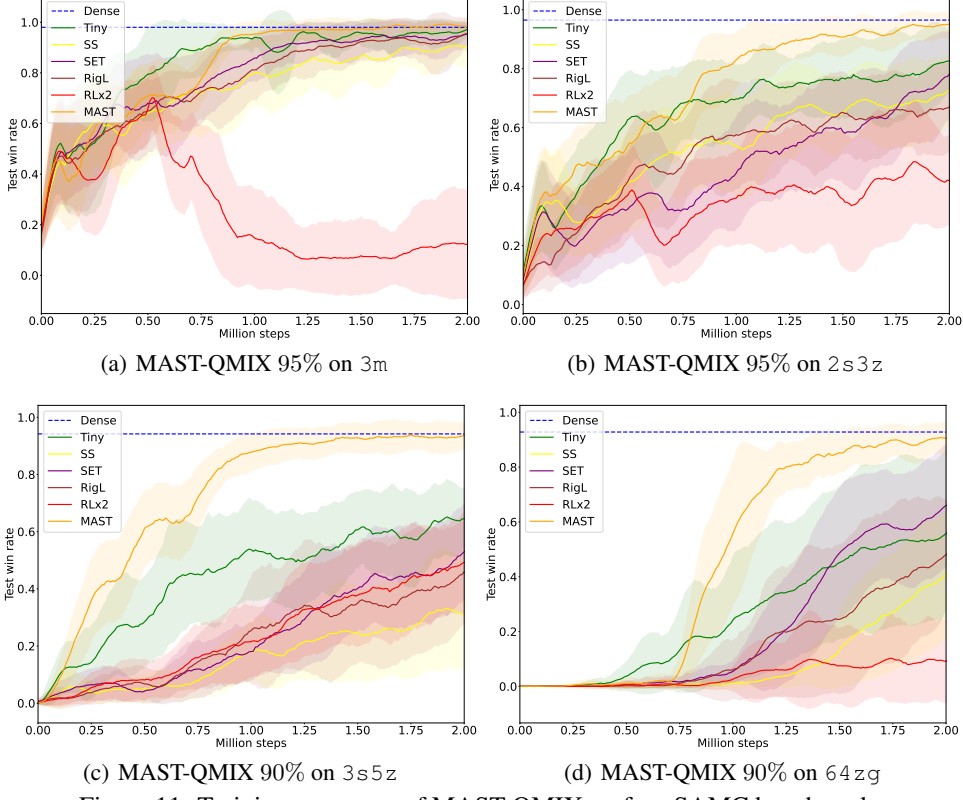

(a) MAST-QMIX 95% on `3m`

(b) MAST-QMIX 95% on `2s3z`

(c) MAST-QMIX 90% on `3s5z`

(d) MAST-QMIX 90% on `64zg`

Figure 11: Training processes of MAST-QMIX on four SAMC benchmarks.

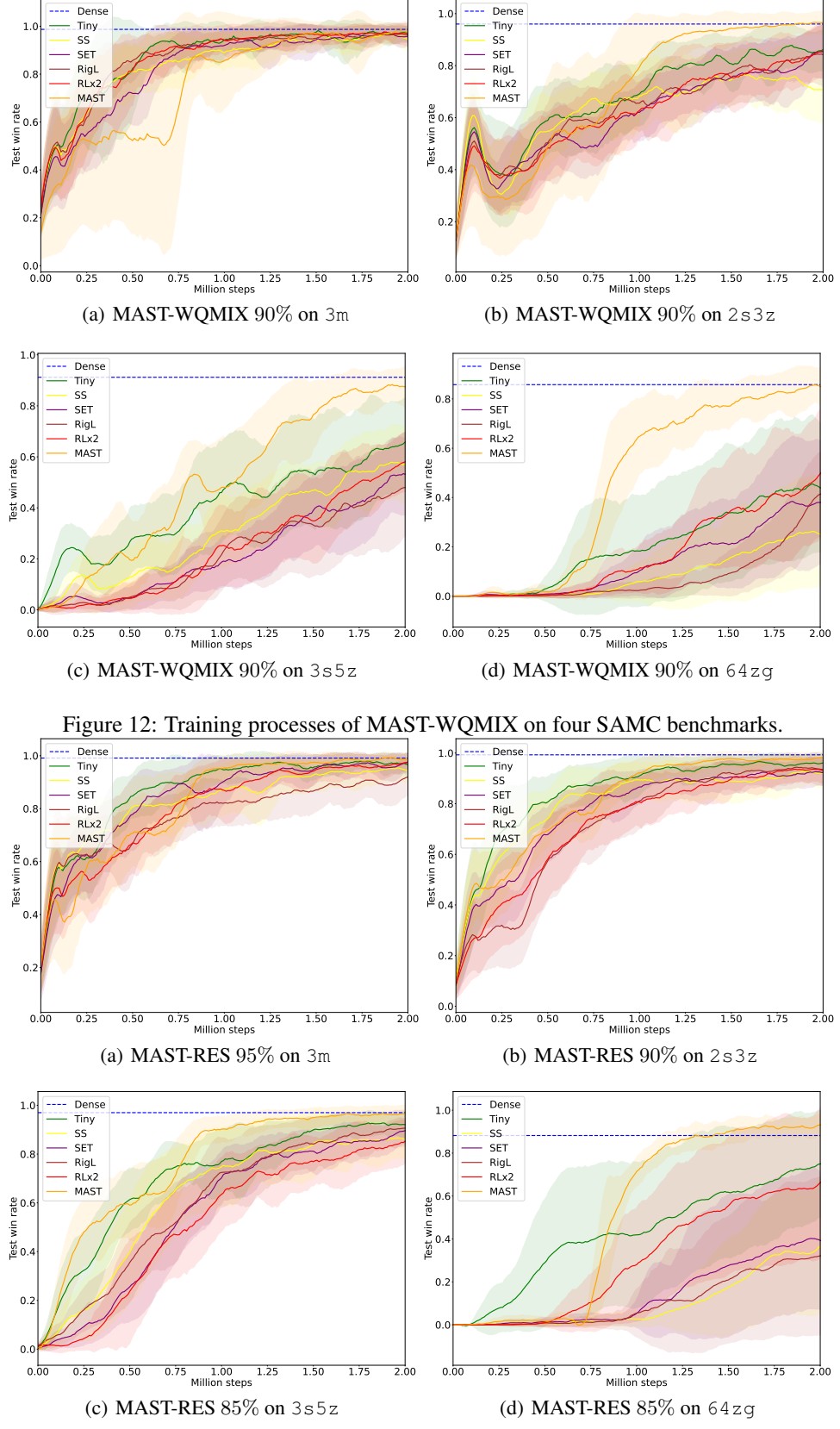

(a) MAST-WQMIX 90% on `3m`

(b) MAST-WQMIX 90% on `2s3z`

(c) MAST-WQMIX 90% on `3s5z`

(d) MAST-WQMIX 90% on `64zg`

Figure 12: Training processes of MAST-WQMIX on four SAMC benchmarks.

(a) MAST-RES 95% on `3m`

(b) MAST-RES 90% on `2s3z`

(c) MAST-RES 85% on `3s5z`

(d) MAST-RES 85% on `64zg`

Figure 13: Training processes of MAST-RES on four SAMC benchmarks.

## B.6  STANDARD DEVIATIONS OF RESULTS IN TABLE 1

Table 7 showcases algorithm performance across four SMAC environments along with their corresponding standard deviations. It's important to note that the data in Table 7 is not normalized concerning the dense model. Notably, MAST's utilization of topology evolution doesn't yield increased variance in results, demonstrating consistent performance across multiple random seeds.

Table 7: Results in Table 1 with standard deviations

| Alg. | Env. | Tiny(%) | SS(%) | SET(%) | RigL(%) | RLx2(%) | Ours(%) |
|------|------|---------|-------|--------|---------|---------|---------|
| Q-MIX | 3m | 96.3±4.3 | 89.8±7.9 | 94.1±5.9 | 93.4±9.5 | 11.9±20.5 | **98.9**±2.0 |
| | 2s3z | 80.8±12.9 | 70.4±13.1 | 74.9±16.4 | 65.8±14.5 | 44.2±17.0 | **94.6**±4.6 |
| | 3s5z | 64.2±11.8 | 32.0±20.3 | 49.3±16.6 | 42.6±19.2 | 47.2±16.2 | **93.3**±5.1 |
| | 64* | 54.0±29.9 | 37.3±23.4 | 62.3±21.9 | 45.2±23.4 | 9.2±15.0 | **89.5**±6.2 |
| | Avg. | 73.8±14.7 | 57.4±16.2 | 70.1±15.2 | 61.7±16.6 | 28.1±17.2 | **94.1**±4.5 |
| WQ-MIX | 3m | 97.0±4.0 | 95.6±4.0 | 96.5±3.6 | 96.5±3.6 | 96.7±4.3 | **97.3**±4.0 |
| | 2s3z | 86.0±7.9 | 72.4±12.4 | 82.5±10.9 | 83.3±10.3 | 83.8±9.9 | **96.2**±4.2 |
| | 3s5z | 64.5±17.9 | 57.0±14.5 | 51.1±15.0 | 46.0±20.5 | 55.4±11.3 | **87.6**±6.9 |
| | 64* | 43.8±27.4 | 25.4±22.0 | 37.8±26.2 | 35.2±16.7 | 45.3±24.7 | **84.4**±8.4 |
| | Avg. | 68.5±13.5 | 62.2±13.0 | 64.0±13.5 | 65.8±11.4 | 70.3±12.5 | **91.4**±5.9 |
| RES | 3m | 96.9±4.1 | 94.7±4.8 | 96.4±4.3 | 90.3±7.4 | 97.0±3.8 | **102.2**±3.2 |
| | 2s3z | 95.8±3.8 | 92.2±5.9 | 92.2±5.5 | 94.0±5.7 | 93.4±5.5 | **97.7**±2.6 |
| | 3s5z | 92.2±4.8 | 86.3±8.8 | 87.6±5.9 | 90.0±7.3 | 83.6±9.2 | **96.4**±3.4 |
| | 64* | 73.5±25.8 | 34.5±29.6 | 38.9±32.3 | 31.1±36.2 | 64.1±33.8 | **92.5**±4.9 |
| | Avg. | 89.6±9.6 | 76.9±12.3 | 78.8±12.0 | 76.3±14.1 | 84.5±13.1 | **97.2**±3.5 |
| Avg. | | 78.7±12.8 | 65.6±13.9 | 72.0±13.7 | 67.8±14.5 | 61.0±14.3 | **94.2**±4.6 |

## B.7  SENSITIVITY ANALYSIS FOR HYPERPARAMETERS

Table 8 shows the performance with different mask update intervals (denoted as $\Delta_m$) in different environments, which reveals several key observations:

- Findings indicate that a small $\Delta_m$ negatively impacts performance, as frequent mask adjustments may prematurely drop critical connections before their weights are adequately updated by the optimizer.

- Overall, A moderate $\Delta_m = 200$ episodes performs well in different algorithms.

Table 8: Sensitivity analysis on mask update interval.

| Alg. | $\Delta_m = 20$ episodes | $\Delta_m = 100$ episodes | $\Delta_m = 200$ episodes | $\Delta_m = 1000$ episodes | $\Delta_m = 2000$ episodes |
|------|------|------|------|------|------|
| QMIX/RES | 99.4% | 97.7% | 99.0% | **100.6%** | **100.6%** |
| WQMIX | 83.2% | 91.9% | **96.1%** | 68.1% | 71.5% |
| Average | 91.3% | 94.8% | **97.5%** | 84.3% | 86.0% |

## B.8  VISUALIZATION OF SPARSE MASKS

We present a series of visualizations capturing the evolution of masks within network layers during the MAST-QMIX training in the 3s5z scenario. These figures, specifically Figure 15 (a detailed view of Figure 10), Figure 17, and Figure 20, offer intriguing insights. Additionally, we provide connection counts for input and output dimensions in each sparse mask, highlighting pruned dimensions. To facilitate a clearer perspective on connection distributions, we sort dimensions based on the descending order of nonzero connections, focusing on the distribution rather than specific dimension ordering. The connection counts associated with Figure 10 in the main paper s given in Figure 14.

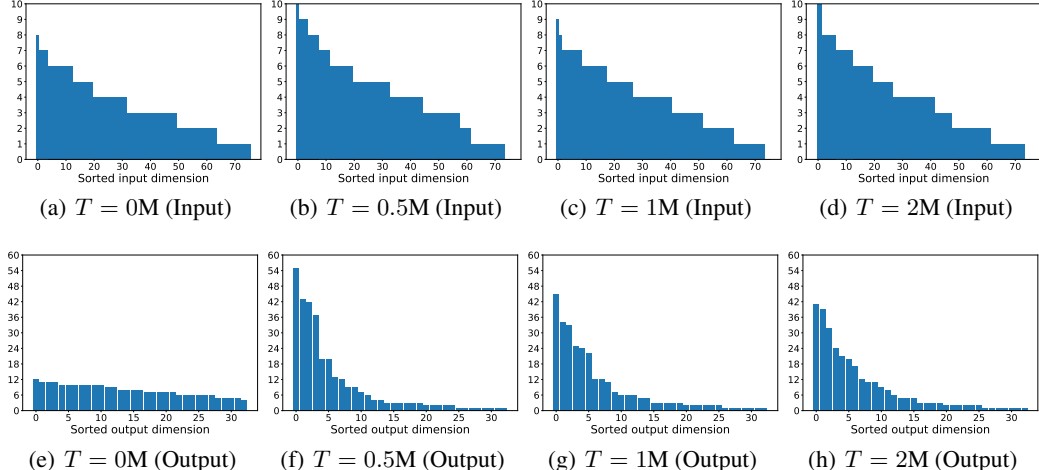

(a) $T = 0$M (Input)  (b) $T = 0.5$M (Input)  (c) $T = 1$M (Input)  (d) $T = 2$M (Input)

(e) $T = 0$M (Output)  (f) $T = 0.5$M (Output)  (g) $T = 1$M (Output)  (h) $T = 2$M (Output)

Figure 14: Number of nonzero connections for input and output dimensions in descending order of the sparse layer visualized in Figure 10.

During the initial phases of training, a noticeable shift in the mask configuration becomes evident, signifying a dynamic restructuring process. As the training progresses, connections within the hidden layers gradually coalesce into a subset of neurons. This intriguing phenomenon underscores the distinct roles assumed by individual neurons in the representation process, thereby accentuating the significant redundancy prevalent in dense models.

- Figure 15 provides insights into the input layer, revealing that certain output dimensions can be omitted while preserving the necessity of each input dimension.

- Figure 17 showcases analogous observations, reinforcing the idea that only a subset of output neurons is indispensable, even within the hidden layer of the GRU.

- Figure 20 presents distinct findings, shedding light on the potential redundancy of certain input dimensions in learning the hyperparameters within the hypernetwork.

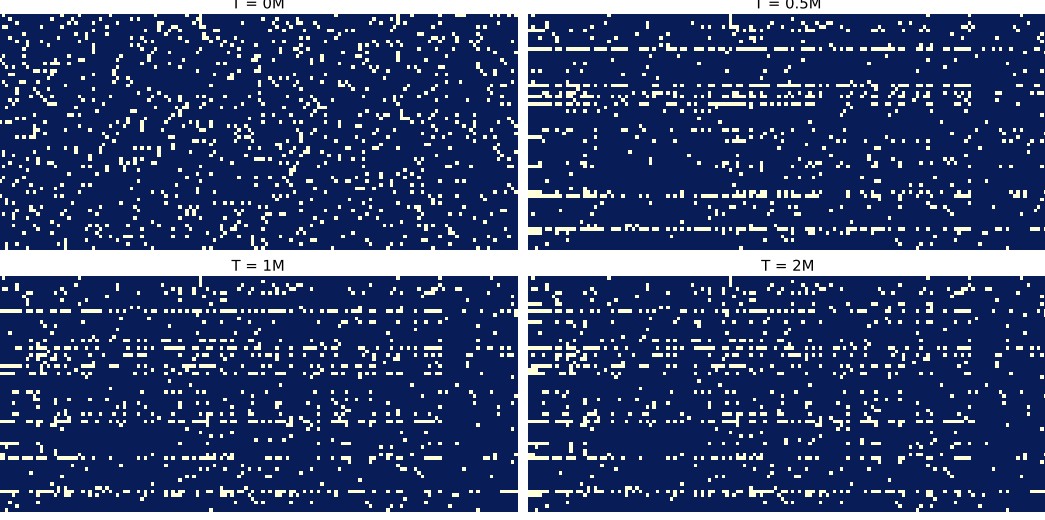

Figure 15: The learned mask of the input layer weight of $Q_1$. Light pixels in row $i$ and column $j$ indicate the existence of the connection for input dimension $j$ and output dimension $i$, while the dark pixel represents the empty connection.

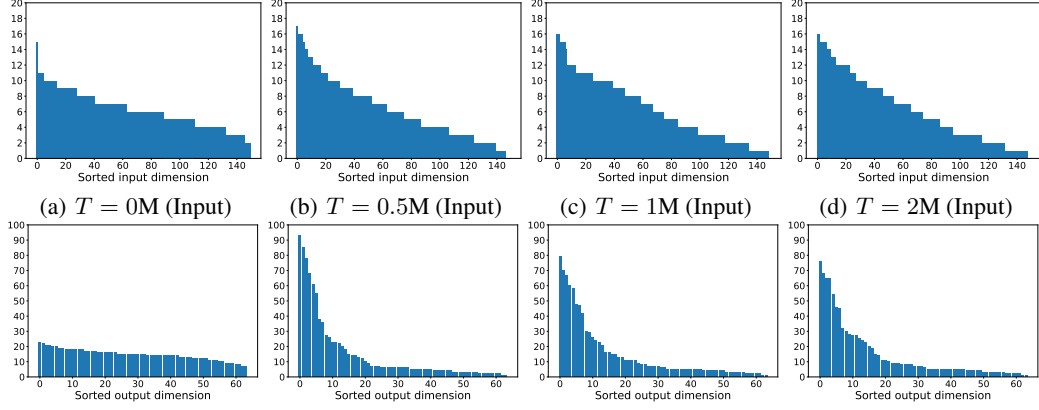

(a) $T = 0M$ (Input) (b) $T = 0.5M$ (Input) (c) $T = 1M$ (Input) (d) $T = 2M$ (Input)

(e) $T = 0M$ (Output) (f) $T = 0.5M$ (Output) (g) $T = 1M$ (Output) (h) $T = 2M$ (Output)

Figure 16: Number of nonzero connections for input and output dimensions in descending order of the sparse layer visualized in Figure 15.

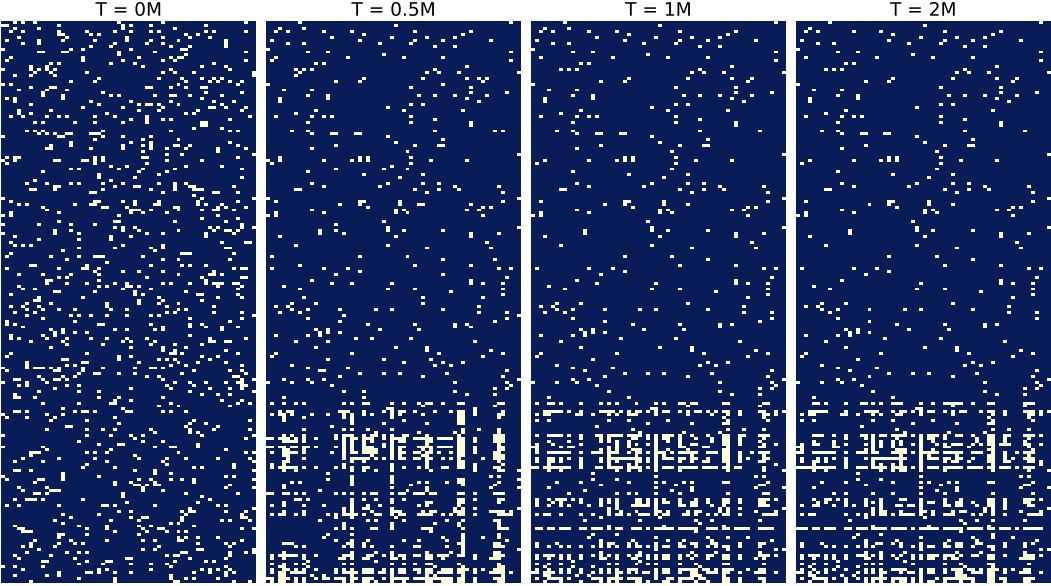

Figure 17: The learned mask of the GRU layer weight of $Q_1$. Light pixels in row $i$ and column $j$ indicate the existence of the connection for input dimension $j$ and output dimension $i$, while the dark pixel represents the empty connection.

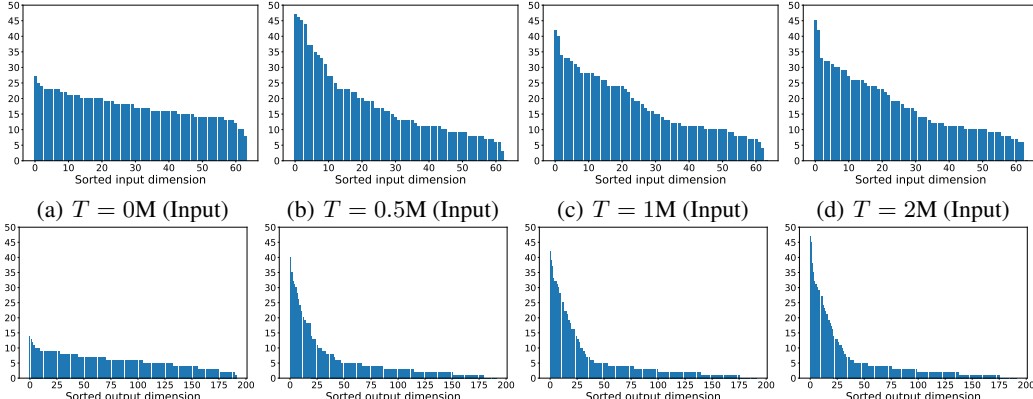

(a) $T = 0M$ (Input) (b) $T = 0.5M$ (Input) (c) $T = 1M$ (Input) (d) $T = 2M$ (Input)

(e) $T = 0M$ (Output) (f) $T = 0.5M$ (Output) (g) $T = 1M$ (Output) (h) $T = 2M$ (Output)

Figure 18: Number of nonzero connections for input and output dimensions in descending order of the sparse layer visualized in Figure 17.

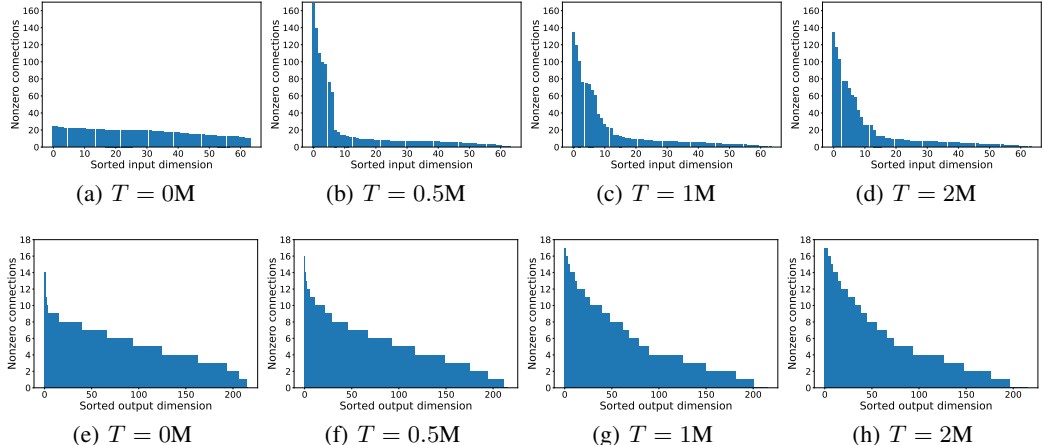

(a) $T = 0M$     (b) $T = 0.5M$     (c) $T = 1M$     (d) $T = 2M$

(e) $T = 0M$     (f) $T = 0.5M$     (g) $T = 1M$     (h) $T = 2M$

Figure 19: Number of nonzero connections for input and output dimensions in descending order of the sparse layer visualized in Figure 20.

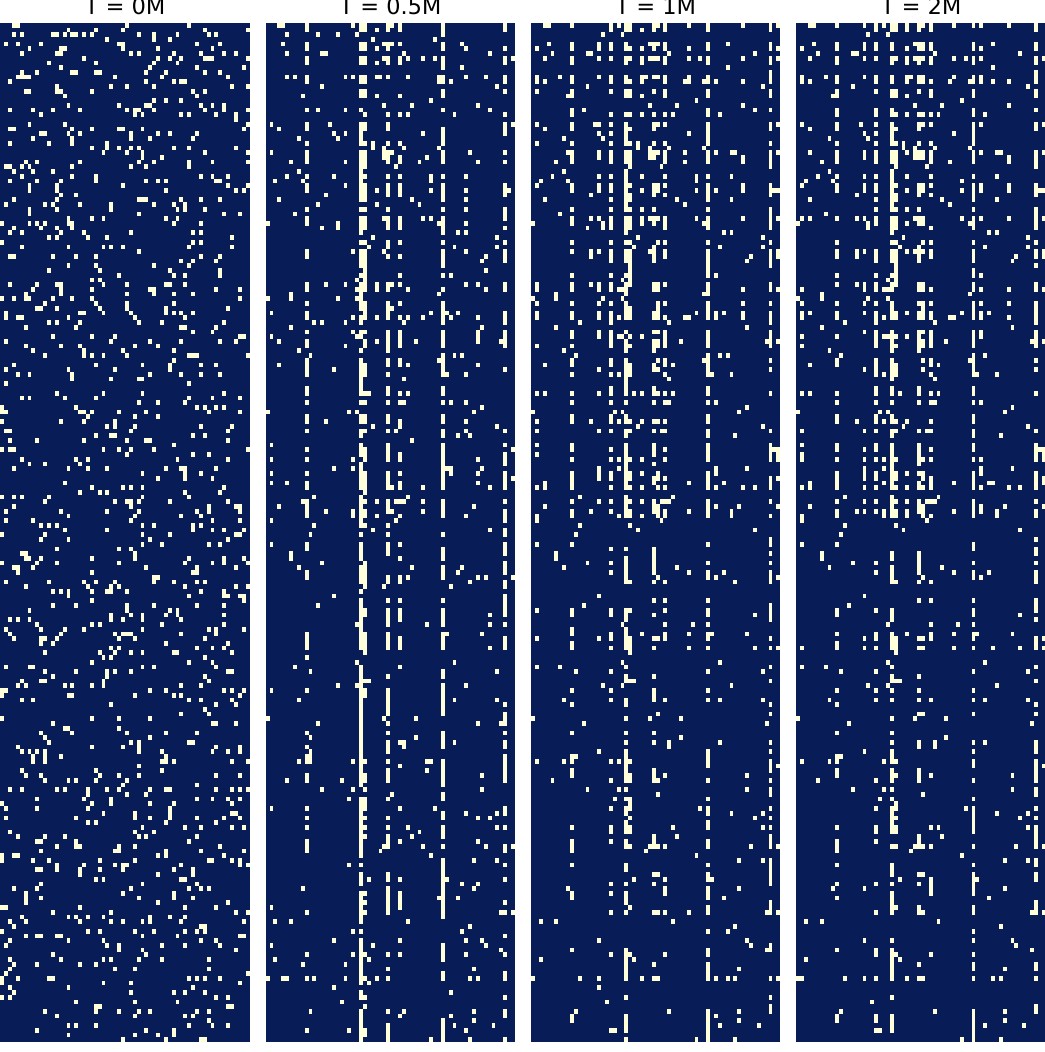

Figure 20: The learned mask of the first layer weight of Hypernetwork. Light pixels in row $i$ and column $j$ indicate the existence of the connection for input dimension $j$ and output dimension $i$, while the dark pixel represents the empty connection.

