# OpenReview forum: "MAST: A Sparse Training Framework for Multi-agent Reinforcement Learning"
_ICLR.cc/2024/Conference — Submitted to ICLR 2024_

### Official Review · Reviewer_ehLP · 2023-10-26

**Soundness:** 3 good
**Presentation:** 4 excellent
**Contribution:** 3 good
**Rating:** 6
**Confidence:** 4

**Summary:**

The paper introduces the Multi-Agent Sparse Training (MAST) framework, which aims to expedite training and enable model compression in MARL. MAST utilizes gradient-based topology evolution to train multiple agents using sparse networks, incorporating a hybrid TD-lambda schema and the Soft Mellowmax Operator to establish reliable learning targets in sparse scenarios. Experiments on the SMAC benchmarks demonstrate the effectiveness of the proposed method.

**Strengths:**

1.	The proposed sparse training framework contributes to making MARL systems applicable to resource-limited devices.
2.	MAST can be applied to different methods with the CTDE training framework.
3.	Experiments on the SMAC benchmarks provide evidence of the effectiveness of the proposed

**Weaknesses:**

1.	The paper utilizes multiple technologies, such as RigL, hybrid TD targets, the Soft Mellowmax operator, and dual buffers, which may make it difficult to discern the specific kernel contribution and novelty.
2.	If the motivation lies in the algorithm, the contribution may seem incremental. Additionally, if the paper aims to design an effective framework, it would be necessary to conduct experiments on other benchmarks, such as Google Research Football, to demonstrate its superiority.

**Questions:**

1.	I would like the authors to clarify their main contribution to help me better understand the paper.
2.	Some curves in the results do not appear to converge at the end. Could this be due to the figures being drawn with smooth weight?
3.	The results were obtained with 4 random seeds. Could you provide information about the variance? Is the method stable?
4.	The paper utilizes many technologies, such as hybrid TD-lambda, which introduces several hyperparameters. How do you decide on these hyperparameters in different scenarios, especially in real-world applications? Do you have any suggestions?

---

> ### Author Response · Authors · 2023-11-19
> **Response to Reviewer ehLP (Part 1 of 2)**
>
> **Thank you sincerely for your invaluable feedback! Below, we provide a detailed response addressing your comments.**
>
> ---
> ### Weaknesses
> ---
>
> > W1: The paper utilizes multiple technologies, such as RigL, hybrid TD targets, the Soft Mellowmax operator, and dual buffers, which may make it difficult to discern the specific kernel contribution and novelty.
>
> We would like to highlight our contribution in three aspects.
> 1. **Originality:**
>    - Addressing the costliness of training and inference in MARL algorithms, especially on resource-limited devices, our MAST framework stands as the *first* algorithm framework achieving 90% sparsity across multiple MARL algorithms. This marks pioneering research in sparse training for MARL.
> 2. **Technical Contributions (Value Learning Enhancements):** Recognizing the inadequacies of sparse MARL models in value learning (indicated in Figure 1), we focused on improving the reliability of training data tagets and training data distributions.
>     - Notably, we introduced $TD(\lambda)$ targets (Section 4.1) to mitigate sparsification errors and addressed overestimation issues in sparse models using the computationally efficient Soft Mellowmax operator (Section 4.2).
>     - In addition, we proposed a dual buffer mechanism to enhance training stability under sparse models by introducing an additional on-policy buffer.
> 3. **Soundness:** Extensive experiments across popular MARL algorithms validate MAST's pioneering role in sparse training:
>     - MAST achieves model compression ratios ranging from 5× to 20× with minimal performance trade-offs (typically under 3%).
>     - Remarkably reduces FLOPs required for training and inference by up to 20×.
>
>
> > W2: If the motivation lies in the algorithm, the contribution may seem incremental. Additionally, if the paper aims to design an effective framework, it would be necessary to conduct experiments on other benchmarks, such as Google Research Football, to demonstrate its superiority.
>
> - **Innovative Framework Design:** Our MAST framework stands as a pioneering effort, marking the first algorithm framework to achieve 90% sparsity across multiple MARL algorithms. This represents groundbreaking research in sparse training for MARL.
> - **Experimental Scope:** Our current experiments predominantly focus on SMAC, given the suitability of our base algorithms (QMIX, WQIX, and RES) within this environment, as previously validated in their original papers (Rashid et al. 2020b; Rashid et al. 2020a; Pan et al 2021). Acknowledging the significance of expanding our experimental scope, we aim to include a broader range of algorithms and diverse environments, such as Google Research Football, in our future work.

---

> ### Author Response · Authors · 2023-11-19
> **Response to Reviewer ehLP (Part 2 of 2)**
>
> ### Questions
> ---
>
> > Q1: I would like the authors to clarify their main contribution to help me better understand the paper.
>
> Please refer to the answer for W1.
>
> > Q2: Some curves in the results do not appear to converge at the end. Could this be due to the figures being drawn with smooth weight?
>
> In our comparative analysis of different algorithms, we ensure a fair assessment by evaluating their performances over a consistent period of interaction with the environment—specifically, 2 million steps, as prior works such as QMIX (Rashid et al. 2020b), WQMIX (Rashid et al. 2020a), and RES (Pan et al. 2021). It's important to note that some curves in the results may not converge towards the end due to the figures being plotted with smoothed results. In some instances, divergent curves might be attributed to particular seeds leading to poorer outcomes. Additionally, certain experiments might not converge at the designated 2 million steps; these cases involve significantly longer training times, resulting in larger FLOPs, making them incomparable with our MAST framework.
>
>
> > Q3: The results were obtained with 4 random seeds. Could you provide information about the variance? Is the method stable?
>
> In our revised experiments, we have expanded our seed set by incorporating an additional four seeds. Consequently, the reported results in Table 1 now reflect the average outcomes derived from a comprehensive set of eight seeds. Moreover, we've included detailed standard deviation information in Table 7 within Appendix B.6, focusing specifically on the win rate presented in Table 1. Besides, the standard deviation information is also depicted through the width of shaded areas within the training curves in Appendix B.5. These inclusive representations serve to underscore the stability and consistency of our MAST framework across various scenarios, showcasing its robust performance.
>
>
> > Q4: The paper utilizes many technologies, such as hybrid TD-lambda, which introduces several hyperparameters. How do you decide on these hyperparameters in different scenarios, especially in real-world applications? Do you have any suggestions?
>
> Our approach employs a consistent set of recommended hyperparameters (detailed in Table 4) across different algorithms and environments, consistently showcasing superior performance of our MAST framework compared to others.
> - For sensitive hyperparameters like $T_0$ and $\lambda$ in hybrid TD($\lambda$) targets and sample partition, we suggest the following values: $T_0=3/8T$, $\lambda=0.6/0.8$, and a sample partition of $5:3$, where $T$ represents the maximum training step.
> - Regarding insensitive hyperparameters such as $\alpha$ and $\omega$ for the Soft Mellowmax operator, we recommend utilizing $\alpha=1$ and $\omega=10$. Our algorithm framework demonstrates insensitivity to these parameters, maintaining performances consistently over 85% compared to the original dense performance, as evidenced in our Section 5.2 ablation experiments.
>
> ---
> **In conclusion, we extend our gratitude to the reviewer for their invaluable feedback. Should our response effectively address your concerns, we kindly hope that the reviewer could consider raising the score rating for our work. Furthermore, we remain available to address any additional queries or points for discussion.**

---

> ### Author Response · Authors · 2023-11-20
> **Reminder to Reviewer ehLP**
>
> Dear reviewer,
>
> Thank you for your effort in reviewing our paper!
>
> We wonder whether our reply fully addresses your concerns. If so, could you please consider raising your score for our work? Please let us know if you have any further questions. We will be more than happy to discuss this with you and answer any remaining questions.
>
> Thank you very much!

---

> > ### Author Response · Authors · 2023-11-21
> > **Another Reminder to Reviewer ehLP**
> >
> > Dear reviewer,
> >
> > Your thoughts on our revisions would be immensely valuable to us as we strive to address all concerns raised during the review process. We're eager to ensure that our paper meets the high standards set by your expertise and the conference's guidelines.
> >
> > We greatly appreciate your time and consideration in reviewing our updated submission. Your prompt response would be sincerely appreciated.

---

> > > ### Comment · Reviewer_ehLP · 2023-11-22
> > > **Thanks for your response.**
> > >
> > > Thank you for your response. I appreciate that most of my concerns have been addressed. I have thoroughly reviewed all the opinions expressed in the other reviews. While the experiments conducted so far have shown promise, the main weakness still lies in the lack of novelty, and I believe further testing in diverse scenarios is necessary. Consequently, I will raise my score to a 6.

---

> > > > ### Author Response · Authors · 2023-11-23
> > > > **Appreciation for your support**
> > > >
> > > > Dear Reviewer ehLP,
> > > >
> > > > Thank you for taking the time to reassess our paper and for your efforts in adjusting the score to 6. Your feedback and guidance have been incredibly valuable to us.
> > > >
> > > > We'd like to emphasize the novelty of our work in training ultra-sparse MARL agents from scratch by focusing on improving the training data target and distribution. Our innovative solution, the MAST framework, pioneers a novel approach that addresses critical challenges in achieving ultra-sparsity while maintaining performance levels, which we believe contributes significantly to the field of sparse MARL.
> > > >
> > > > We acknowledge your concern about the experiment scope and wish to assure you that we are committed to expanding our experimentation by incorporating more algorithms and diverse environments into our MAST framework. Your recognition of our work would greatly enhance its visibility and contribute significantly to its recognition within our field.
> > > >
> > > > Once again, we sincerely appreciate your valuable feedback and your ongoing support of our work.

---

### Official Review · Reviewer_JDXo · 2023-10-30

**Soundness:** 2 fair
**Presentation:** 1 poor
**Contribution:** 2 fair
**Rating:** 3
**Confidence:** 4

**Summary:**

This paper introduces MAST, a novel sparse training framework for deep MARL, utilizing gradientbased topology evolution to efficiently explore network configurations in sparse models. MARL faces significant challenges in ultra-sparse models, including value estimation errors and training instability. Their experiments show the contribution on FLOPs and performance.

**Strengths:**

1.The problem that the authors focus on is very important and valuable to explore.
2.A lot of experiments have been conducted to prove their contribution.

**Weaknesses:**

1.The writing logic is bad, making readers hard to follow. For example, what is the relationship of the sparse model in SL, single-agent RL and multi-agent RL? Why does MASK apply to QMix series approaches and when MAST is applied to QMIX series algorithms and leverage the RigL method for topology evolution? The authors use too many words on the related work and basic knowledge of MARL, but not clarify the logic clearly.
2.Some of the formulas are not numbered.
3.Cannot the discount rate in RL be 1?
4.The experiment is conducted only on 4 seeds, which is not enough and strong in RL scenarios.
5.“These topology adjustments occur infrequently throughout the training process, happening every 200 episodes (about 10,000 steps) in our specific configuration.” How about under other configurations?
6.Algorithm 1 of the overall procedure is in supplementary, I suggest to contain it in the main text.

**Questions:**

Please see the weakness above. Can the authors give a clear logic of the paper? And the innovative solutions in an easy-understood way?

---

> ### Author Response · Authors · 2023-11-19
> **Response to Reviewer JDXo (Part 1 of 2)**
>
> **Thank you sincerely for your invaluable feedback! Below, we provide a detailed response addressing your comments.**
>
> ---
> ### Weaknesses
> ---
> > W1. The writing logic is bad, making readers hard to follow. For example, what is the relationship of the sparse model in SL, single-agent RL and multi-agent RL? Why does MASK apply to QMix series approaches and when MAST is applied to QMIX series algorithms and leverage the RigL method for topology evolution? The authors use too many words on the related work and basic knowledge of MARL, but not clarify the logic clearly.
>
> We've integrated the following bullet points in our revised manuscript and would greatly appreciate any specific areas that still require further clarification:
>
> 1. **Relationship of Different Sparse Models**
>     - Sparse networks, initially proposed in **deep supervised learning**, alongside state-of-the-art sparse training frameworks such as SET (Mocanu et al., 2018) and RigL (Evci et al., 2020), can train a 90%-sparse network without performance degradation from scratch.
>     - In **DRL**, the learning target evolves in a bootstrap way (Tesauro et al., 1995), and the training data distribution can be non-stationary (Desai et al., 2019). Existing works (Sokar et al., 2021; Graesser et al., 2022) reveal that directly adopting a DST method in **single-agent RL** fails to uniformly achieve good compression across different environments. RLx2 (Tan et al. 2022) marks the first successful sparse training throughout, surpassing 90% sparsity.
>     - As depicted in Figure 1, neither DST nor RLx2 works in **multi-agent RL**. The complexities in **multi-agent RL**, such as inaccurate value estimation and training instability, are addressed by our MAST framework. MAST utilizes a novel hybrid TD(λ) target mechanism, coupled with the Soft Mellowmax operator, which facilitates precise value estimation even in the face of extreme sparsity. Additionally, MAST unveils a dual buffer mechanism designed to bolster training stability in sparse environments.
>
> 2. **Why QMIX?**
>     - Our focus lies on algorithms adhering to the Centralized Training with Decentralized Execution (CTDE) paradigm (Oliehoek et al., 2008; Kraemer & Banerjee, 2016). Within this paradigm, QMIX (Rashid et al., 2020b) stands as a representative algorithm.
>
> 3. **Related Work and Basic Knowledge of MARL**
>     - We've condensed and restructured the related work and preliminary sections in our revision.
>
> *References:*
> (Tesauro et al., 1995) Gerald Tesauro et al. Temporal difference learning and td-gammon. Communications of the ACM, 38(3):58–68, 1995.
> (Desai et al., 2019) Shrey Desai, Hongyuan Zhan, and Ahmed Aly. Evaluating lottery tickets under distributional shifts. EMNLP-IJCNLP 2019, pp. 153, 2019.
>
> > W2: Some of the formulas are not numbered.
>
> We have numbered all formulas in our revision.
>
> > W3: Cannot the discount rate in RL be 1?
>
> We consider the case that discount rate is less than 1.
> * On the one hand, most Reinforcement Learning algorithms (such as SARSA or Q-learning) and Dynamic Programming algorithms converge solely towards the optimal policy under the discounted reward infinite horizon criteria.
> * On the other hand, when the discount rate is below 1, our $TD(\lambda)$ targets play a significant role in mitigating the network fitting error resulting from network sparsification, as discussed in Section 4.1.
>
> > W4: The experiment is conducted only on 4 seeds, which is not enough and strong in RL scenarios.
>
> In our revised experiments, we have expanded our seed set to include an additional four seeds. Consequently, the results reported in Table 1 now represent the average outcomes derived from a total of eight seeds.
>
> > W5: “These topology adjustments occur infrequently throughout the training process, happening every 200 episodes (about 10,000 steps) in our specific configuration.” How about under other configurations?
>
> In our revision, we've conducted an ablation study focusing on the mask update interval ($\Delta_m$), detailed in Table 8 within Appendix B.7. Analysis from Table 8 reveals several key observations:
>
> - Findings indicate that a small $\Delta_m$ negatively impacts performance, as frequent mask adjustments may prematurely drop critical connections before their weights are adequately updated by the optimizer.
> - Overall, A moderate $\Delta_m=200$ episodes performs well in different algorithms.
>
> > W6: Algorithm 1 of the overall procedure is in supplementary, I suggest to contain it in the main text.
>
> In our revised version, Algorithm 1 has been relocated to Section 4, accompanied by additional explanations detailing our topology evolution scheme.

---

> ### Author Response · Authors · 2023-11-19
> **Response to Reviewer JDXo (Part 2 of 2)**
>
> ### Questions
> ---
> > Q1: Can the authors give a clear logic of the paper? And the innovative solutions in an easy-understood way?
>
> **Summary of Paper Logic:**
>
> 1. **Issues of Dynamic Sparse Training (DST) in MARL:** DST, stemming from supervised learning, holds potential for accelerating training and compressing models in MARL. However, direct adoption of DST methods in MARL fails to maintain performance at high sparsity levels, as indicated in Figure 1. This points to the challenge of sparse MARL models struggling with value learning, a crucial aspect for policy learning.
>
> 2. **Improving Value Learning in Sparse MARL Models:** MAST introduces innovative solutions to address the accuracy of value learning in ultra-sparse models by concurrently refining training data targets and distributions.
>    - **Enhancing Learning Targets:** Recognizing larger network fitting errors in sparse models, we introduce $TD(\lambda)$ targets (Section 4.1) to mitigate sparsification errors. Additionally, to address overestimation issues in sparse models, we employ the computationally efficient Soft Mellowmax operator (Section 4.2).
>    - **Stabilizing Training Data:** Observing training instability in MARL algorithms under sparse models with a single off-policy buffer, our proposed dual buffer mechanism (Section 4.3) improves stability by introducing an extra on-policy buffer.
>
> 3. **An Innovative Solution (MAST) with Empirical Validation:**
>    - Integrating these enhancements, our innovative solution, MAST, achieves superior sparsity levels exceeding 90% without performance degradation in MARL, validated through our experiments. This work represents pioneering research in sparse training for MARL.
>
> We have also updated our paper accordingly. Please see our revision.
>
> ---
> **In conclusion, we extend our gratitude to the reviewer for their invaluable feedback. Should our response effectively address your concerns, we kindly hope that the reviewer could consider raising the score rating for our work. Furthermore, we remain available to address any additional queries or points for discussion.**

---

> ### Author Response · Authors · 2023-11-20
> **Reminder to Reviewer JDXo**
>
> Dear reviewer,
>
> Thank you for your effort in reviewing our paper!
>
> We wonder whether our reply fully addresses your concerns. If so, could you please consider raising your score for our work? Please let us know if you have any further questions. We will be more than happy to discuss this with you and answer any remaining questions.
>
> Thank you very much!

---

> > ### Author Response · Authors · 2023-11-21
> > **Another Reminder to Reviewer JDXo**
> >
> > Dear reviewer,
> >
> > Your thoughts on our revisions would be immensely valuable to us as we strive to address all concerns raised during the review process. We're eager to ensure that our paper meets the high standards set by your expertise and the conference's guidelines.
> >
> > We greatly appreciate your time and consideration in reviewing our updated submission. Your prompt response would be sincerely appreciated.

---

> > > ### Comment · Reviewer_JDXo · 2023-11-22
> > > **Reply to the authors**
> > >
> > > Thanks for the authors' response. After reading other reviewers' concerns, I will remain my score. There are lots of meaningful points in the paper, but please revise the paper carefully and add some of the replies to the main paper. The current paper is below the acceptance.

---

> > > > ### Author Response · Authors · 2023-11-22
> > > > **Gratitude for Your Feedback and Seeking Further Suggestions**
> > > >
> > > > Dear Reviewer JDXo,
> > > >
> > > > We deeply appreciate your insightful feedback and suggestions on our paper. We've diligently revised the manuscript based on your valuable recommendations, *encompassing reorganized paper logic, a comprehensive elucidation of various sparse models' relationships, incorporated formula numbers, conducted additional experiments with four extra seeds, and included an additional ablation study on the mask update interval.*
> > > >
> > > > Your expertise is immensely valuable to us, and we are committed to further improving our work. We would greatly appreciate it if you could kindly point out any **specific areas** where you believe further enhancements could be made to elevate the paper's quality and contribution.
> > > >
> > > > Thank you once again for your time, support, and continued engagement with our work.

---

### Official Review · Reviewer_LxMW · 2023-11-04

**Soundness:** 3 good
**Presentation:** 2 fair
**Contribution:** 3 good
**Rating:** 5
**Confidence:** 4

**Summary:**

This paper involves sparse training for MARL to reduce the computation cost. Besides, to reduce the value estimation error, a hybrid TD($\lambda$) and Soft Mellowmax operator are incorporated. Experiments on SMAC show the proposed method significantly reduces the training cost while maintains good performance.

**Strengths:**

Using sparse training in MARL is relatively new and an important direction that will inspire the community.

Experiments are conducted on SMAC with extensive analysis.

**Weaknesses:**

The clarity of this paper needs to be improved. For example, the proposed method uses RigL to sparse the network. However, the details of RigL are missing, which makes it confusing for readers who are not familiar with the sparse training area.

The limitation of this paper is not discussed. For example, there are too many key parameters that need to be fine-tuned, making it infeasible to apply to other complex domains.

The literature review lacks some closely related work, such as [1]. So the statement 'The only existing endeavor to train sparse MARL agents' is inaccurate. Also, dual buffers have some related work like [2].

The visualization does not look very informative to the reviewer, as there are no specific patterns for the latent space distribution. Perhaps projecting what connections are removed and what connections are remaining and analyzing why it is like that will be interesting.

[1] Parameter Sharing with Network Pruning for Scalable Multi-Agent Deep Reinforcement Learning. AAMAS 2023.

[2] PMIC: Improving Multi-Agent Reinforcement Learning with Progressive Mutual Information Collaboration. ICML 2022.

**Questions:**

Please see the pros and cons part.

Could you explain more about why 'larger values under a sparse model compared to a dense network' in Section 4.1? Do you mean overestimations?

The well-known method to deal with overestimation is double Q-learning, have you compared this with SM? Which one is better and why?

How do you select the value of $\omega$ as 5 and 10?

Why does the value in Table 1 exceed 100%? How do you calculate it?

The common evaluation metric in SMAC is average success rate, why do you use average reward? Do you normalize all tasks' rewards to the same scale?

---

> ### Author Response · Authors · 2023-11-19
> **Response to Reviewer LxMW (Part 1 of 2)**
>
> **Thank you sincerely for your invaluable feedback! Below, we provide a detailed response addressing your comments.**
>
> ---
> ### Weaknesses
> ---
> > W1: The clarity of this paper needs to be improved. For example, the proposed method uses RigL to sparse the network. However, the details of RigL are missing, which makes it confusing for readers who are not familiar with the sparse training area.
>
> We've incorporated additional details about RigL in Section 4 in our revision. We would greatly appreciate any specific areas that still require further clarification.
>
> > W2: The limitation of this paper is not discussed. For example, there are too many key parameters that need to be fine-tuned, making it infeasible to apply to other complex domains.
>
> In our revision, we've discussed the limitations concerning implementation and hyperparameter tuning in Appendix A.4. Additionally, we also provide Table 4 in Appendix B.3 for practical recommendations for both sensitive and insensitive hyperparameters across different algorithms and environments:
>
> 1. **For sensitive hyperparameters** (such as $T_0$ and $\lambda$ for hybrid TD($\lambda$) targets and sample partition), we recommend values of $T_0=3/8T$, $\lambda=0.6/0.8$, and a ratio of $5:3$ for the sample partition, where $T$ represents the maximum training step.
>
> 2. **For insensitive hyperparameters** (such as $\alpha$ and $\omega$ for the Soft Mellowmax operator), flexibility exists in their settings. However, we recommend following our suggestion of $\alpha=1$ and $\omega=10$. Our algorithm framework demonstrates insensitivity to these parameters, with performance consistently exceeding 85% compared to the original dense performance, as illustrated in Table 3.
>
> *Notably*, as depicted in Table 1, employing the recommended hyperparameter settings (as detailed above) across different algorithms and environments consistently positions our MAST as the top performer.
>
>
> > W3: The literature review lacks some closely related work, such as [1]. So the statement 'The only existing endeavor to train sparse MARL agents' is inaccurate. Also, dual buffers have some related work like [2].
>
> We have included the related work mentioned in our revision.
>
> > W4: The visualization does not look very informative to the reviewer, as there are no specific patterns for the latent space distribution. Perhaps projecting what connections are removed and what connections are remaining and analyzing why it is like that will be interesting.
>
> In our revision, we've provided connection counts for both input and output dimensions in Figure 10 of the main paper and Appendix B.8 for further insight into topology evolution. These new illustrations specifically highlight the non-zero connection counts in both input and output dimensions during the training, shedding light on pruned dimensions. Surprisingly, our findings reveal that a small portion of neurons in both input and output layers prove necessary, while others exhibit minimal or no connections after MAST training. This observation underscores the substantial redundancy present in the original dense network and the efficacy of our MAST framework in removing such redundancy. Notably, this observation aligns with the sparse model obtained in single-agent RL, as demonstrated in Appendix C.11 of (Tan et al., 2022).

---

> ### Author Response · Authors · 2023-11-19
> **Response to Reviewer LxMW (Part 2 of 2)**
>
> ### Questions
> ---
> > Q1: Could you explain more about why 'larger values under a sparse model compared to a dense network' in Section 4.1? Do you mean overestimations?
>
> This sentence was amended in our revision as follows: *"Denote the network fitting error as $\epsilon(s,\boldsymbol{u})=Q_\text{tot}(s,\boldsymbol{u};{\theta})-Q_\text{tot}^{\pi_t}(s,\boldsymbol{u})$, it will be larger under an improper sparsified model compared to a dense network, as evidenced in Figure 1 where improper sparsified models fail in learning good policy."* This statement highlights an empirical observation, implying that the network's performance might significantly deteriorate when trained with an improperly sparsified neural network.
>
> As indicated in Figure 1, the use of static sparse networks or classical dynamic sparse training methods detrimentally affects network performance, leading to an increase in the network fitting error. This effect occurs when the network is only sparsified without other modifications in data distribution or data targets, causing a notable rise in the absolute value of the network fitting error.
>
>
> > Q2: The well-known method to deal with overestimation is double Q-learning, have you compared this with SM? Which one is better and why?
>
>
> Indeed, the algorithm we experimented with, such as QMIX, incorporates measures to mitigate overestimation issues, notably employing double Q-learning (utilizing $Q$ network and target $Q$ network for each agent's value esimation in QMIX). However, as demonstrated in Figure 4, relying solely on double Q-learning remains insufficient in resolving the overestimation challenges within sparse models. Our experimental findings indicate that despite the inclusion of double Q-learning in QMIX, the overestimation issue persists, and our SM operator significantly contributes to addressing this limitation within the framework.
>
> > Q3: How do you select the value of w as 5 and 10?
>
> We adhere to the recommendations outlined in prior research on the Soft Mellowmax operator, specifically as detailed in Gan et al. (2021). On the other hand, we note that MAST's performances are insensitivity to these parameters ($\alpha$ and $\omega$), with performance consistently exceeding 85% compared to the original dense performance, as illustrated in Table 3.
>
> > Q4: Why does the value in Table 1 exceed 100%? How do you calculate it?
>
> As highlighted in the caption of Table 1, all data is normalized with respect to the dense model. The performance attained by MAST surpasses that of the original dense model when the value exceeds 100%.
>
> > Q5: The common evaluation metric in SMAC is average success rate, why do you use average reward? Do you normalize all tasks' rewards to the same scale?
>
> This was a typo, and we've corrected it in our revised version. Our primary evaluation metric is the average win rate. In Table 1 in the main paper, the performance, i.e. the win rate, of different algorithms is normalized with respect to the dense model. We provide a comprehensive report on the exact average win rate in Table 7 in the Appendix B.6 with standard deviation.
>
> ---
> **In conclusion, we extend our gratitude to the reviewer for their invaluable feedback. Should our response effectively address your concerns, we kindly hope that the reviewer could consider raising the score rating for our work. Furthermore, we remain available to address any additional queries or points for discussion.**

---

> ### Author Response · Authors · 2023-11-20
> **Reminder to Reviewer LxMW**
>
> Dear reviewer,
>
> Thank you for your effort in reviewing our paper!
>
> We wonder whether our reply fully addresses your concerns. If so, could you please consider raising your score for our work? Please let us know if you have any further questions. We will be more than happy to discuss this with you and answer any remaining questions.
>
> Thank you very much!

---

> ### Author Response · Authors · 2023-11-21
> **Another Reminder to Reviewer LxMW**
>
> Dear reviewer,
>
> Your thoughts on our revisions would be immensely valuable to us as we strive to address all concerns raised during the review process. We're eager to ensure that our paper meets the high standards set by your expertise and the conference's guidelines.
>
> We greatly appreciate your time and consideration in reviewing our updated submission. Your prompt response would be sincerely appreciated.

---

> > ### Comment · Reviewer_LxMW · 2023-11-21
> > **ACK**
> >
> > Thanks for the response. I already have all the information I need for the discussion phase. Thanks.

---

> > > ### Author Response · Authors · 2023-11-22
> > > **Cordial Request for Reconsideration**
> > >
> > > Dear Reviewer LxMW,
> > >
> > > Thank you for dedicating your time to review our response to your insightful comments on our paper. Your feedback has been instrumental in enhancing the quality and depth of our work.
> > >
> > > We are sincerely grateful for your engagement and would like to respectfully request your reconsideration of the paper rating. We have diligently addressed the key points highlighted in your review, incorporating additional details about RigL in Section 4, a comprehensive discussion on limitations in Appendix A.4, providing practical recommendations for key hyperparameters in MAST through Table 4, enriching the related work section, and including supplementary visualization results on sparse masks. These revisions, we believe, have substantially elevated the paper's quality and contribution.
> > >
> > > Your acknowledgement of the revisions made by adjusting the rating would significantly bolster our paper's visibility and recognition within our field.

---

### Author Response · Authors · 2023-11-19
**Revision Highlights**

We thank all reviewers for insightful comments and helpful suggestions. We have revised the paper accordingly, with revised parts marked in blue.

- We've expanded on RigL details in Section 4 to elucidate the topology evolution step, aiding clearer comprehension. Besides, Section 4 underwent logic updates to enhance readability and facilitate a better understanding of our contributions.
- We've conducted additional experiments by expanding our seed set with an additional four seeds. Table 1 now represents the average outcomes from a total of eight seeds. Additionally, an ablation study on the mask update interval ($\Delta_m$) has been performed.
- In Appendix A.4, we've discussed the limitations and future prospects of our work, particularly regarding implementation and hyperparameter tuning, demonstrating self-criticism.
- Both a comprehensive related work section in Appendix A.1 and a concise version in Section 2 of the main paper have been included.
- For deeper insight into topology evolution, connection counts for sparse layers' input and output dimensions are presented in Section 5.3 of the main paper and Appendix B.8.
- Practical hyperparameter recommendations are now available in Table 4 within Appendix B.3, catering to deploying the MAST framework in scenarios beyond our experiment scope.

---

### Meta-Review · Area_Chair_SC6K · 2023-12-04

**Metareview:**

This paper introduces a novel Multi-Agent Sparse Training (MAST) framework that capitalizes on gradient-based topology evolution to exclusively train multiple MARL agents using sparse networks.

**Reviewers have reported the following strengths:**

- Novelty and significance of the considered problem;
- Experimental evaluation;
- Generalizability of the proposed method.

**Reviewers have reported the following weaknesses:**

- Quality of writing;
- Incremental contribution;
- Doubts on technicalities.

**Decision**

The Reviewers' opinion about the novelty of the paper is somewhat different. However, I deem the novelty of this paper sufficient enough. However, the paper still lacks of clarity, which contributes to the confusion about the novelty of this work. Importantly, the authors' rebuttal has not been sufficient to change the Reviewers' opinion about this work. I strongly encourage the authors improving the presentation of their work for a future resubmission.

**Justification For Why Not Higher Score:**

N/A

**Justification For Why Not Lower Score:**

N/A

---

### Decision · Program_Chairs · 2024-01-16

Reject